



# The ALADIN System and its Canonical Model Configurations AROME CY41T1 and ALARO CY40T1

Piet Termonia[1,2], Claude Fischer[3], Eric Bazile[3], François Bouyssel[3], Radmila Brožková[4], Pierre Bénard[3], Bogdan Bochenek[5], Daan Degrauwe[1,2], Maria Derkova[6], Ryad El Khatib[3], Rafiq Hamdi[1], Ján Mašek[4], Patricia Pottier[3], Neva Pristov[7], Yann Seity[3], Petra Smolíková[4], Oldrich Spaniel[6], Martina Tudor[8], Yong Wang[9], Christoph Wittmann[9], and Alain Joly[3]

[1]Royal Meteorological Institute, Brussels, Belgium
[2]Department of Physics and Astronomy, Ghent university, Ghent, Belgium
[3]Météo-France, Toulouse, France
[4]Czech Hydrometeorological Institute, Prague, Czech Republic
[5]Institute of Meteorology and Water Management - State Research Institute of Poland, Poland
[6]Slovak Hydrometeorological Institute, Bratislava, Slovakia
[7]Slovenian Environment Agency, Ljubljana, Slovenia
[8]Meteorological and Hydrological Service, Zagreb, Croatia
[9]Zentralanstalt für Meteorologie und Geodynamik, Vienna, Austria

*Correspondence to:* Piet Termonia (piet.termonia@meteo.be)

**Abstract.**

The ALADIN System is a numerical weather prediction system (NWP) developed by the international ALADIN consortium for operational weather forecasting and research purposes. It is based on a code that is shared with the global model IFS of the ECMWF and the ARPEGE model of Météo-

France. Today, this system can be used to provide a multitude of high-resolution limited-area model (LAM) configurations. A few configurations are thoroughly validated and prepared to be used for the operational weather forecasting in the 16 Partner Institutes of this consortium. These configurations are called the ALADIN Canonical Model Configurations (CMCs). There are currently three CMCs: the ALADIN baseline-CMC, the AROME CMC and the ALARO CMC. Other configurations are

possible for research, such as process studies and climate simulations.

The purpose of this paper is (i) to define the ALADIN System in relation to the global counterparts IFS and ARPEGE, (ii) to explain the notion of the CMCs and to document their most recent versions, and (iii) to illustrate the process of the validation and the porting of these configurations to the operational forecast suites of the Partner Institutes of the ALADIN consortium.

This paper is restricted to the forecast model only; data assimilation techniques and postprocessing techniques are part of the ALADIN System but they are not discussed here.



## 1 Introduction

The ALADIN System[1] [2] is the set of pre-processing, data assimilation, forecast model and post-processing/verification software codes shared and developed by the Partners of the ALADIN con-
sortium to be used for running a high-resolution limited-area model (LAM) for producing the best possible operational Numerical Weather Prediction (NWP) applications based on a configuration compatible with their available computing resources. The ALADIN consortium is a collaboration between the National (Hydro)Meteorological Services (NHMSs) of 16 European and North-African countries[3], see ALADIN international team (1997). This consortium was created in 1990. It carries
out an ambitious research program and has delivered a state-of-the-art NWP system that is used by its Members states for their operational weather-forecasting applications.

The collaboration follows the initial objectives of the consortium, as they were introduced by its founder Jean-François Geleyn:

(a) to have or to gain with the help of other members the capability to define, build and run local
versions of the ALADIN System, but also,

(b) to build the capability to conceive, develop, test and ultimately integrate scientific ideas locally and finally in the new versions of the ALADIN System.

Both objectives lead to the benefit of all through the exchange of expertise and the improvements of the ALADIN System, and contributes to the steady progress of the discipline of NWP (Bauer et al.,
2015). One consequence is that the consortium as a whole is responsible for the code as a whole. Therefore, creating a new version of the source code and its maintenance is a transversal activity within the consortium.

While all Partner services have the capacity to implement their operational versions of the AL-ADIN System by themselves, some activities are organized into more formally structured coopera-
tions to develop applications that go beyond the deliverables of the ALADIN consortium.

The ALADIN consortium hosts the geographically localized Regional Cooperation for Limited-Area Modelling in Central Europe consortium (RC LACE), with seven members: the national National (Hydro-)Meteorological Services of Austria, Croatia, Czech, Hungary, Romania, Slovakia

---

[1] The ALADIN acronym stands for Aire Limitée Adaptation Dynamique Développement International (International development for limited-area dynamical adaptation)

[2] See http://www.umr-cnrm.fr/aladin/.

[3] Currently the Partners of the ALADIN consortium are (1) Office National de la Météorologie, Algeria, (2) Zentralanstalt für Meteorologie und Geodynamik, Austria, (3) Royal Meteorological Institute of Belgium, Belgium, (4) Bulgarian National Institute of Meteorology and Hydrology, Bulgaria, (5) Meteorological and Hydrological Service, Croatia (6) Czech Hydrometeorological Institute, Czech Republic, (7) Météo-France, France, (8) Hungarian Meteorological Service, Hungary, (9) Direction de la Météorologie Nationale, Morocco, (10) Institute of Meteorology and Water Management - State Research Institute of Poland, Poland, (11) Instituto Português do Mar e da Atmosfera, Portugal, (12) National Meteorological Administration of Romania, Romania, (13) Slovak Hydrometeorological Institute, Slovakia, (14) Slovenian Environment Agency, Slovenia, (15) Institut National de la Météorologie de Tunisie, and (16) Turkish State Meteorological Service, Turkey.





and Slovenia. It contributes a lot on the development of the ALADIN System. It made key contri-
butions to the ALADIN non-hydrostatic dynamical core and the development of the physics param-
eterizations, in particular the ALARO CMC that will be described in section 3.3. This consortium
provides extra resources to exchange and to process meteorological data used for the operational
data assimilation systems in the RC LACE Partner countries. RC LACE develops and operates a
pan-European probabilistic system Limited Area Ensemble Forecasting LAEF based on the AL-
ADIN System (Wang et al., 2010, 2011, 2012, 2014; Weidle et al., 2013, 2016; Bellus et al., 2016;
Schellander-Gorgas et al., 2017)

Since 2005, the ALADIN consortium also shares its code with the HIRLAM consortium[4] through
a cooperation agreement (Bengtsson et al., 2017).

The codes of the ALADIN System are common with the codes of the global Integrated Fore-
cast System (IFS) of the ECMWF[5] and the global ARPEGE model[6] of Météo-France (Courtier
and Geleyn, 1988; Courtier et al., 1991). The common, shared codes of the ALADIN System are
managed in a central repository maintained by Météo-France with the help of the Partners of the
ALADIN consortium. From this repository versions of the ALADIN System are assembled on a
regular basis following the updates of the IFS cycles and the scientific improvements developed
within the LAM community. This includes an assembling of the latest developments of ECMWF
and Météo-France. The code evolution of the ALADIN System is thereby triggered by (i) updates
with respect to IFS/ARPEGE versions, (ii) the implementation of novel scientific developments
and (iii) specific code modernization (e.g. towards object-oriented code design) or optimization (for
High-Performance Computing, HPC).

The aim of this link between the LAM and global models is threefold. First we can consider the
configurations of the ALADIN System as limited-area configurations of the global model. Secondly,
by sharing parts of the codes, the maintenance efforts can be reduced and developments done in
either global or limited-area models become mutually available. Lastly, as mentioned by (Warner
et al., 1997), keeping a maximum of consistency between the global model and the LAM model
dynamics and physics can reduce the errors at the lateral boundaries (LBCs) and can be beneficial
for the lateral-boundary coupling of the LAM.

A quasi infinite number of choices can be made in the scientific physics and dynamics options of
the configurations of the ALADIN System. This offers a high degree of freedom for the participating
Partners of the ALADIN consortium to configure their national NWP applications, and even to
develop tailor-made applications to address specific requests from their end users. On the other
hand, it should be stressed that not all combinations of the available dynamics and physics schemes
lead to scientifically meaningful model configurations.

---

[4]HIgh-Resolution Limited-Area Model consortium
[5]European Centre for Medium-Range Weather Forecasts
[6]Action de Recherche Petite Echelle Grande Echelle



Historically the ALADIN model was created as the LAM version of ARPEGE (Radnóti et al., 1995). Since all of the ALADIN countries nowadays target their applications at resolutions within the so-called convection permitting scales (1-5 km), two physically-consistent model configurations called AROME[7] (Seity et al., 2011) and ALARO[8] have been developed to address the need for applications at these resolutions. The current efforts to assemble, validate, document and maintain new versions of the ALADIN System, are focused on these two 'canonical' model configurations. However, in order to keep the close link with the global model ARPEGE, a LAM configuration that uses the ARPEGE physics is maintained. This configuration is still called the ALADIN model configuration. The new versions of these ALADIN model configurations are not collectively exported to operational NWP applications of the ALADIN Partners anymore, but they undergo a minimal validation and can be used in scientific projects where a mesoscale model is needed.

The purpose of the present paper is,

1. to articulate the link between the LAM configurations of the ALADIN System and the global models IFS and ARPEGE;

2. to present a comprehensive description of the ALADIN System, including the notion of Canonical Models Configurations (CMCs). Many aspects of the ALADIN System have been published in the literature, a systematic overview is provided here, while citing the papers whenever available, and adding some lacking descriptions to provide a complete picture;

3. to provide a status review of the current scientific content of the CMCs that are based on the recent cycles CY40T1 and CY41T1, and to illustrate how both of them are linked and implemented in the model code;

4. to illustrate the development and validation aspects for the latest version of the ALADIN System. This paper does not give a full overview of the implementations in the 16 member states of the ALADIN consortium, but it will present a few cases as an illustration of the model validation.

The scope of this paper will be limited to the forecast model configurations. For instance, data assimilation (Fischer et al., 2005; Wattrelot et al., 2014; Brousseau et al., 2016) is part of the ALADIN System codes but will not be described here nor any postprocessing methods.

While the consortium activities are focused on numerical weather prediction, some configurations of the ALADIN System have also been used for climate simulations. This will not be discussed in the present paper either. For a few examples of such applications, see for instance Déqué et al. (1994), De Troch et al. (2013) and Giot et al. (2016).

This paper is organized as follows. In section 2 the ALADIN System will be described. The purpose is to define the ALADIN System by indicating its specificities related to code architecture with

---

[7]AROME stands for Application of Research to Operations at Mesoscale.
[8]ALARO stands for ALadin-AROme.



respect to the global models ARPEGE and IFS, paying special attention to the validation process of the newest version of the ALADIN System. In section 3 the notion of CMCs will be introduced in more detail. The scientific description of the recent CMCs will be presented. Section 4 will illustrate how the recent versions have been exported to the ALADIN Partner countries. The paper will be concluded with a discussion and a short outlook in section 5.

## 2 Description of the ALADIN System

### 2.1 Generalities

A Version of the ALADIN System is a release of the ALADIN System. Some Versions are distributed at regular times to the ALADIN Partners for research and development, as well as for operational purposes. These Versions are called export versions. A Configuration of the ALADIN System is a subset of ALADIN Codes used by a consortium member for its own implementation. Canonical Model Configurations (CMCs) are configurations of the ALADIN System for which the ALADIN consortium organizes collective efforts for the scientific and technical validation according to the state of the art of the latest research and development. The consortium also organizes the coordination and networking activities in order to install and run these canonical configurations in the operational NWP suites of the ALADIN Consortium Members.

Today there are two CMCs in the full sense: the AROME model configuration and the ALARO model configuration. While the ALADIN configuration is not exported to the Partners of the consortium anymore, it is considered as the baseline-CMC to ensure the link with the global model ARPEGE.

The scientific developments of the ALADIN System are implemented in a five-step process. The consortium carries out joint research and development activities with the aim of maintaining the ALADIN System at scientific and technical state of the art level within the NWP community. It carries out the necessary scientific and technical studies to define and maintain the ALADIN System and its Canonical Model Configurations. The consortium organizes the general maintenance of the ALADIN System with the aim to create new Versions on a regular basis. It organizes coordination and networking activities in order to support the ALADIN Consortium members in their ability to run the ALADIN Canonical Model Configurations on the computing platforms of their national Institutes. The consortium provides a platform for sharing scientific results, numerical codes, operational environments, related expertise and know-how, as necessary for all ALADIN Consortium members to conduct operational and research activities with the same tools.



Code updates are done about every 6 months: one common with IFS/ARPEGE, one common only to the ALADIN Partners.

A new Version build is planned about one year in advance, and this original kick-off decision is followed by an "upstream coordination" process with the intention to anticipate as much as possible any potential conflict between expected code commitments. This effort is considered strategic for the NWP system, due to its highly integrated nature, and it is involving scientific experts along with system (programming) experts.

The practical steps of the initial build of a new ALADIN Version release are mostly taking place at Météo-France: merge of code contributions, early validation process. Progressively, as the early versions become technically stable, some remote installation and further validation can take place, until the new release is declared. This process does not comprise pre-operational local implementations in which then the meteorological quality of a new release is evaluated, beyond the technical

tests.

The technical validation is done in several steps, some of which being ignored if found unnecessary:

1. a benchmark of base tests: adiabatic model versions, change of model grid geometry versions, tangent-linear/adjoint model run tests, and specific forecast tests including physics packages

among which those used for defining the CMCs;

2. comparison with the previous reference version, aiming to trace back changes that disrupt bit reproducibility, or to put it differently, verifying that bit reproducibility is broken for understood reasons;

3. computation of statistical scores such as bias and root-mean-square errors (RMSE) with re-

spect to observations or reference analyses;

4. specific model output diagnostics used in research mode like averages of model tendencies;

5. one-dimensional model tests to assess profiles of fields and their tendencies;

6. specific data assimilation test periods are run (the time period is chosen in order to match with a recent context for the throughput of observations).

This process is meant to bring the embedded implementations of the LAM configurations of the ALADIN System in phase with the cycles of the global IFS and the ARPEGE models and is called "phasing". The cycle numbers of the ALADIN Versions are the same as the corresponding cycles of IFS and ARPEGE. The outcome of the build and validation process is a new Version of the ALADIN System labelled in the Météo-France central source code repository. Mature Versions of

the ALADIN System are packages in so-called "export versions" for installation in the ALADIN Partner centers.





### 2.2 The scientific and technical specificities of the code architecture of the ALADIN System

The definition of the ALADIN System is rooted in the options of the shared code to configure the LAM model configurations. This section describes the architecture of the code to outline what is common with the global model and what differentiates the LAM configurations from the global model.

One of the main concerns in the developments of these codes is the special care taken to be able to run the model configurations with long time steps or, to put it non-dimensionally, with large Courant numbers. Most of the choices in the development of the numerical treatments of the dynamics and the physics parameterizations are made from that point of view. As far as is known today, from recent intercomparisons (see e.g. Michalakes et al., 2015) this key feature, combined with dual parallelization capabilities makes IFS/ARPEGE/ALADIN models the most efficient or cheapest ones to run, each in their categories, in terms of "time to solution".

The code of the ALADIN System is shared with the code of the IFS of ECMWF and the ARPEGE model of Météo-France. The current operational versions use a spectral dynamical core with a two-time level semi-Lagrangian semi-implicit scheme (Ritchie et al., 1995; Robert et al., 1972; Simmons et al., 1978; Temperton et al., 2001). The use of a spectral transform method naturally implies that there is no horizontal staggering of the variables in the gridpoint calculations part. To solve the semi-implicit problem, the dynamic equations are reduced to a single Helmholz equation in the horizontal divergence, see Caluwaerts et al. (2015, 2016). In the equations of the dynamics the $u$ and $v$ components of the wind fields are recast in terms of absolute momentum. As such the Coriolis term, as well as the curvature terms, do not appear on the right-hand side and, as a result, do not enter the linearized semi-implicit (SI) formulation. Indeed, the approach taken to solve the SI problem is remarkably efficient insofar as the problem is horizontally separable: then, the spectral method enables an elegant, direct purely algebraic solution. This efficiency is lost whenever parameters depending on the horizontal coordinates are kept in the linear problem. Actually, one such parameter, the map factor, does enter the SI problem, but its horizontal dependency is handled in a semi-analytical way, leading to a weakly non-diagonal problem in spectral space, therefore enabling to keep most of the advantages of the spectral solving method.

The time-step computations are organized in such a way that the same dynamics formulations can be used for both limited-area and global geometries. The time-step algorithm is schematically outlined in table 1 in a simplified manner. Mind that this algorithm is not the same for IFS as far as the physical parameterizations calculations are concerned. In the IFS, the physics is performed on variables at different times depending on the physical process, whereas in the ARPEGE model and the ALADIN System it is performed entirely on the $t - \delta t$ state variable before calling the explicit part of the dynamics, see Termonia and Hamdi (2007).

The code can be run with a non-hydrostatic dynamical core that solves the fully compressible Euler equations (Bubnová et al., 1995). This dynamical core is referred to as ALADIN-NH and may



**Table 1.** Schematic overview of the time-step algorithm of the configurations of the ALADIN System and the choices that differentiate them with respect to the global ARPEGE model.

| | step | options (LAM vs. global) |
|---|---|---|
| 1. | horizontal derivatives (vorticity, divergence and pressure/temperature gradients) | |
| 2. | inverse spectral transform: spectral to gridpoint | bi-FFT$^{-1}$ / Legendre, FFT |
| 3. | computation of the physics contributions | AROME physics / ALADIN/ALARO physics |
| 4. | calculation of the tendencies of the prognostic variables of the model state | INTFLEX |
| 5. | computation of the explicit gridpoint dynamics and adding it to the total tendencies of the prognostic variables | IFS/ARPEGE/ALADIN hydrostatic / ALADIN-NH |
| 6. | computation of the semi-Lagrangian departure points and interpolation of the tendencies to these points | SLHD |
| 7. | addition of the interpolated tendencies to the model state | |
| 8. | lateral boundary coupling | bi-periodic LBC conditions |
| 9. | direct spectral transforms | bi-FFT / Legendre, FFT |
| 10. | solving the semi-implicit Helmholtz equation | IFS/ARPEGE/ALADIN hydrostatic / ALADIN NH |

be used in both AROME CMC and ALARO CMC for horizontal grid resolutions in which non-
hydrostatic effects play an important role, i.e. roughly 3 km, depending on the details of the used
numerical scheme.

The vertical coordinate system uses a mass-based hybrid pressure terrain-following coordinate $\eta$
(Simmons and Burridge, 1981; Laprise, 1992). The vertical discretization is based on finite differ-
ences (Simmons and Burridge, 1981) or finite elements using B-splines of general order
(Vivoda and Smolíková, 2013). Unlike the hydrostatic case, in the ALADIN-NH dynamical core
not only the integral operators but also the vertical derivatives need to be discretized since they ap-
pear in the set of basic equations. Moreover, the basic constraints being satisfied in the continuous
case with the finite-differences vertical discretization are not fulfilled by the finite-element vertical
discretization. It follows that the elimination of all prognostic variables but one is not possible when
solving Helmholtz equation and an iterative procedure is being applied in this case.

There are two additional prognostic variables compared to the hydrostatic model core: the non-
hydrostatic pressure departure from the hydrostatic pressure and a specific expression of the vertical-
divergence variable, denoted as $d$.

This choice ensures satisfactory stability properties of the semi-implicit scheme (Bénard et al.,
2004, 2005). However, in the semi-Lagrangian advection scheme, in the case of a flow over steep
slopes, the accuracy of the calculation may be reduced depending on the choice of the bottom bound-
ary condition for $d$. The solution proposed by Smith (2002) is to use the vertical wind $w$ instead of



vertical divergence in the explicit part of the semi-implicit calculations. This allows the free-slip
lower boundary condition to be introduced in its most natural form, without the need for any ex-
tra assumptions. These simpler calculations then lead to an enhanced accuracy in the vicinity of
steep slopes. Vertical staggering of prognostic variables is a necessary consequence of this approach
resulting in the calculation of two sets of semi-Lagrangian trajectories, one at full model levels for
most of the prognostic variables and a second one at the intermediate levels for the vertical velocities.
Furthermore, a transformation from $w$ to $d$ and vice versa needs to be performed at the beginning
and at the end of the explicit computations. Recently, more conservative semi-Lagrangian horizontal
weights were proposed which take into account the deformation of air parcels along each direction
(Malardel and Ricard, 2015).

The non-hydrostatic equation set can be solved using a separable, linear non-iterative semi-implicit
problem. However, the parameter domain of stability is reduced with respect to the hydrostatic case.
One way of improving it is to use two distinct temperatures in the scheme, instead of a single one.
Roughly, one characterizes gravity waves, the other acoustic waves. To go further, Bénard (2003)
proposes to see the semi-implicit scheme as a highly linearized single iteration approximation to the
tangent-linear iterative fix-point search of the more exact solution. From this analysis, he derives a
more stable but iterative scheme called the Iterative Centered Implicit scheme. A number of dynam-
ical non-linear terms are recomputed at each iteration, with optional precision (and cost) levels, and
the SI solved again with recomputed right-hand terms. This scheme can alternatively be viewed as
belonging to the predictor-corrector family.

The dynamical core (both hydrostatic and non-hydrostatic) includes a linear numerical horizontal
diffusion based on a power of the Laplace operator as proposed by Jakimow et al. (1992). The
operator is included in the solver of the Helmholtz equation in the spectral part of the computations
in step 10 in table 1 and is thus solved implicitly. Fourth order is usually used in the operational
applications. For the iterative centered implicit time scheme, the spectral horizontal diffusion is
applied at each iteration step, whilst physical tendencies and semi-Lagrangian trajectories may not
be recomputed and could be kept from the predictor step.

Additionally the code allows to use the non-linear Semi-Lagrangian Horizontal Diffusion (SLHD)
scheme, computed under step 6 of the time-step algorithm in table 1. The original version of the
scheme was developed and implemented by Váňa et al. (2008). Later its conservative properties
were improved by using a carefully constructed class of semi-Lagrangian interpolators, exploiting
the fact that accuracy and damping properties of an interpolator are not strictly tied. On a 4-point
stencil in one dimension it is possible to construct a class of second order accurate interpolators
with broadly varying damping, and with spectral selectivity equivalent to the fourth order diffusion.
An additional control of spectral response is obtained by using an optional Laplacian smoother.
Non-linearity of the SLHD scheme is achieved via a modulation of the diffusion strength by the
horizontal deformation rate of the flow. Due to its grid-point character, the scheme enables to apply





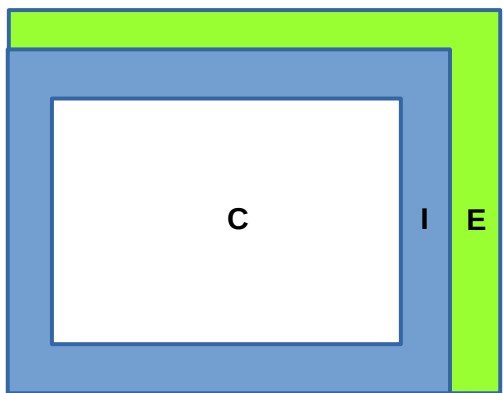

**Figure 1.** The domain of the LAM model is composed of three zones: a physical central zone (C), an interme-
diate zone (I) where the lateral-boundary conditions are imposed by a relaxation, and the extension zone (E)
where artificial periodic extensions of the fields are inserted.

diffusion also on quantities that are not transformed to spectral space, such as specific humidity,
cloud condensates, or the turbulent kinetic energy (TKE).

The shared code also allows to perform a Digital-Filtering Initialization (DFI) on a model state
(Lynch, 1990). In operation applications an optimal version is used (Lynch et al., 1997) based on
a Dolph-Chebyshev filter (Lynch, 1997). Termonia (2008) observed that such temporal filters may
filter out fast moving signals in the small scales and implemented a Scale-Selective Digital-Filtering
Initialization (SSDFI) in the shared ARPEGE/ALADIN code.

Most of the above-described features are embedded in the common code with the global ARPEGE
model. Three features differentiate the ALADIN System configurations from its global counter part:

1. the choice of the horizontal bi-Fourier spectral transform instead of the spherical spectral
transforms (steps 1, 2, 9 in table 1) and a formulation of the Helmholtz equation in term of the
proper operators and map factors (step 10),

2. the lateral-boundary conditions (LBCs) (step 8 in table 1) and

3. the physics packages which are adapted in step 3 in table 1, for an application at the high-
resolutions targeting the convection-permitting scales, as shown in Fig. 2.

The structure of the geographical domain of the LAM configurations is based on the idea of
Haugen and Machenhauer (1993). It has three zones as shown in Fig. 1 consisting of a physical
central zone (C), an intermediate zone (I) where the lateral-boundary conditions are imposed by





a relaxation, and a so-called extension zone (E) where artificial periodic extensions of the fields
are inserted before performing the direct fast Fourier transforms. The double periodicity implies
that the geometry of the spectral LAM is essentially a torus as opposed to a sphere for the global
model configurations. In operational applications the C+I domain is most commonly mapped onto
the sphere by means of a conformal-Lambert projection. The other two conformal projections are
also possible, namely the polar stereographic and the Mercator projections.

The LAM configurations of the ALADIN System use the Davies (1976) relaxation scheme in the
I zone in Fig. 1, which nudges the fields from the fields of the host model to the guest model. Instead
of using the proposed nudging coefficients by Davies (1983), in the ALADIN System this is done
by a parameterized function:

$$\alpha(z) = 1 - (p+1)\, z^p + p z^{p+1}\,, \tag{1}$$

where $z$ is the normalized distance form the boundary of the C zone to the border of the I zone. The
shape of the relaxation curve $\alpha$ is fixed by tuning the variable $p$ (the current configurations use a
value of $p = 2.16$ for wind and temperature, and $p = 5.52$ for water vapor and hydrometeors).

In the ALADIN System the lateral-boundary conditions are imposed in step 8 in table 1 just
before the spectral transforms. This is done by relaxing the result of the explicit part of the dynamics
(computed in step 5 in table 1) to the fields of the host model after they have been subjected to the
operator of the semi-implicit scheme as proposed by Radnóti (1995). Symbolically this looks like,

$$\mathbf{X}^{cpl} = \alpha \mathbf{X}_G + (1-\alpha)\left(1 - \frac{\Delta t}{2}\mathcal{L}\right)\mathbf{X}_H\,, \tag{2}$$

where $\mathbf{X_G}$ is the updated tendency of the LAM model state after step 7, $\mathbf{X}_H$ is the field of the host
model, $\mathcal{L}$ is the linear operator of the semi-implicit scheme and $\alpha$ is taken as in Eq. 1. The result of
Eq. 2, $\mathbf{X}^{cpl}$ is then transformed to spectral space and becomes the input to the Helmholtz solver in
step 10. The fields are made periodic in the extension zone by spline functions.

The new biperiodization and LBC scheme proposed by Boyd (2005) has been implemented in the
ALADIN System by Termonia et al. (2012). They introduced some other options to adapt it to the
semi-Lagrangian scheme and to make the scheme more flexible. For instance, the code can be run
with a disjoint split between the relaxation in the I zone and the biperiodic windowing in the E zone
of Fig. 1, which improves upon the original proposal of Boyd (2005) where the relaxation and the
biperiodic windowing overlap. It has been shown that such a configuration with a truncation of the
semi-Lagrangian trajectories at the edge of the C+I zone, gives better results than the Davies scheme
(Degrauwe et al., 2012).

In practice the configurations of the ALADIN System are coupled to the IFS or to the ARPEGE
model. To this end the dynamical fields are spatially interpolated to the LAM domain. The periodic
extensions are inserted in the E zone at this stage. To run the system with Boyd's scheme, one needs
the information of the fields of the host model outside the C and the I zone, see Termonia et al. (2012).
The results are stored in files. These files usually contain the spectral coefficients of the dynamical



fields. Such files are created at Météo-France or ECMWF and transferred to the ALADIN Partners

in a timely manner. They are computed with the resolution corresponding to the average horizontal
resolution of the driving model, not the target one, to save bandwidth and transfer time. These files
are short-handedly called the telecom files.

The interpolation software also allows to interpolate the fields of a LAM configuration to a LAM
subdomain with possibly a new resolution. The telecom files are created at regular times with one-

hour, three-hour or six-hour time intervals. These files are read during a forecast run of the guest
model and interpolated in time to get the fields at each time step. Mind that time interpolations of the
bi-periodic fields yields bi-periodic fields. In practice the time interpolation is carried out by a linear
interpolation or a quadratic interpolation (Tudor and Termonia, 2010). Termonia (2004) found that a
temporal interpolation of 3-h coupling updates may, in rare cases of a fast moving storm entering the

domain through the boundaries, result in errors of up to about 10 hPa in the mean-sea level pressure
fields (Termonia et al., 2009). Termonia et al. (2011) proposed to use an error-detection procedure
based on a recursive digital filtering procedure within the global model and to apply a restart in such
cases. This procedure is used operationally in the forecast suite of the Royal Meteorological Institute
(RMI). Alternative ways for detecting the errors from the fields available in the telecom files from

IFS have been explored (Tudor, 2015).

The scientific content of the physics schemes that are called under step 3 in table 1 for ALADIN,
ALARO and AROME will be described in section 3.

The coupling of the physics to the dynamics (step 4 in table 1) is based on a flux-conservative
formulation developed by Catry et al. (2007). A flexible version of this physics-dynamics interface,

called INTFLEX, has been recently implemented and validated in the common code by Degrauwe
et al. (2016) that facilitates the implementation of new species and processes. The use of INTFLEX
for the AROME configuration has improved the life-cycle dynamics of the cold pool mechanism
in deep convective systems. The INTFLEX code functions as an interface routine to plug in the
different physics packages in the time-step algorithm. It is common to the ARPEGE model and to

the configurations of the ALADIN System.

For the efficiency of the LAM configurations on modern parallel computing architectures, the
same strategies as for the global IFS/ARPEGE models are employed, with limited needs of adapta-
tion. Mostly thanks to ECMWF and the integration concept, this code is characterized by a rather
rare fully parameterized dual parallelization capability. This means that the code can use various mix

of distributed memory parallel tasks and shared memory parallel threads. On the current dominant
interconnected multi-CPU boards, the LAM configurations primarily use the same cache-blocking
mechanism for cache-based computers[9] (Zwieflhofer et al., 2003; Hamrud et al., 2012). This comes

---

[9] These are the so-called NPROMA blocks, named after the dimensioning NPROMA variable. This variable was initially
designed to optimize the vectorization length on vector machines. The NPROMA blocking was developed first for vector
shared memory machines. Then the code was adapted for vector distributed memory machines by introducing MPI. Then
OpenMP has been progressively implemented.





along with two-dimensional Message Passing distributions (MPI), both in spectral space, and in gridpoint space. On top of this cache-blocking slicing the LAM configurations can further use a parallelism by OPEN-MP threads.

Recently, the performances on large computing domains has been significantly improved by introducing an input/output server developed by Météo-France. It enables to resume the time integration itself, while the writing to disk is performed in parallel. Reading may also be distributed. Dual parallelization makes it possible to use multicore boards. Dual parallelization combined with parallel I/O together with a much reduced number of time-steps to reach a given forecast range makes these codes extremely efficient, even though the transpositions required by the use of spectral transforms are not ideal from a scalability viewpoint.

The main three particularities of the LAM parallelism with respect to the global model configurations concern:

1. the handling of the coupling data in gridpoint space, for which a specific Message Passing distribution and parallelism has been developed;

2. the handling of the limited area aspects in gridpoint space. Unlike in the global model, the semi-Lagrangian trajectories have to be constrained to the physical area C+I and possibly a margin of the extension zone in the case of the Boyd solution mentioned above. Also, the semi-Lagrangian trajectories are computed on a plane, which requires, among other things, to construct the so-called halo for the MPI implementation in a different way.

3. In spectral space, the distributed Fourier-transform code is shared with the global model in the zonal direction; while in the other direction a second distributed Fourier transform code replaces the distributed Legendre transforms.

## 3   The Canonical Model Configurations

The three physics packages ALADIN, AROME and ALARO can be called under step 3 of the time-step organization in table 1. Their target resolutions are illustrated in Fig. 2. The AROME CMC and the ALARO CMC are respectively based on the cycles CY41T1 and CY40T1 and both are described in sections 3.2 and 3.3.

### 3.1   The ALADIN baseline-CMC

The current ALADIN baseline CMC calls the ARPEGE physics that is used at Météo-France between summer 2013 and spring 2017. Here we limit ourselves to a brief description of this version.

Its radiation scheme is based for the long-wave on the so-called RRTM scheme (Mlawer et al., 1997; Iacono et al., 2008) and for the short wave the six-band Fouquart-Morcrette scheme (Fouquart and Bonnel, 1980; Morcrette, 1993). The boundary layer parameterization is based on a prognostic





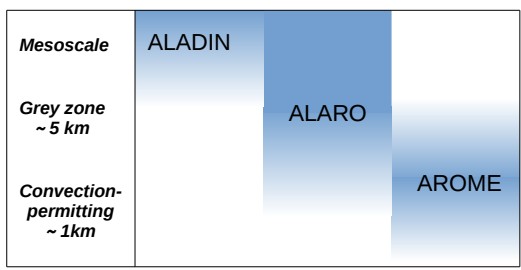

**Figure 2.** The different LAM configurations of the ALADIN System and their target resolutions.

equation of the Turbulent Kinetic Energy (Cuxart et al., 2000) associated with a shallow convection scheme (KFB) based on a CAPE closure (Bechtold et al., 2001), both schemes are linked to the thermal production of TKE computed by the KFB scheme and by a modification of the original mixing length from Bougeault and Lacarrere (1989) by the shallow cloud from KFB (Bazile et al., 2011).

The deep convection is represented by an updated version of the mass-flux scheme based on a moisture convergence closure (Bougeault, 1985). Alternatively, deep convection can now be represented using the PCMT scheme (Prognostic Condensates Microphysics and Transport) (Piriou et al., 2007; Guérémy, 2011). This scheme is already operational in the ARPEGE ensemble prediction system, and will soon be in ARPEGE. The cloud microphysics has four prognostic variables (cloud water

and ice and liquid and solid precipitation) for the resolved precipitation (Lopez, 2002; Bouteloup et al., 2005) and the probability distribution function for the statistical cloud scheme comes from (Smith, 1990). A parameterization of subgrid orographic effects (Catry et al., 2008) represents gravity wave drag, wave deposition, wave trapping, form drag and lift effects. For the continental surface the SURFEX software (Masson et al., 2003) is used with the options used in the AROME model

configuration, as will be described below and in section 3.2.

### 3.2   The AROME CMC

The AROME canonical model configuration has been developed to run in the convection-permitting resolutions starting from 2.5-km resolution. It is a non-hydrostatic convective-scale limited-area model setup described by Seity et al. (2011) and Brousseau et al. (2016). Its physical parameteriza-

tions come mostly from the Méso-NH research model (Lafore et al., 1998) whereas the dynamical core is the Non-Hydrostatic ALADIN one described in section 2.2. It is run with a light, single-iteration predictor-corrector step which allows to use long time steps (50s at 1.3km horizontal resolution for instance). The recent versions of the AROME configurations[10] use the COMAD scheme

---

[10]COMAD is active in the ALADIN System code since CY40T1 and in particular in the current cycle CY41T1 described here.





**Table 2.** The AROME CMC

| parameterization/dynamics | scheme | references |
| --- | --- | --- |
| dynamics | non-hydrostatic ALADIN | Bénard et al. (2010) |
| radiation | RRTMG_LW, SW6 | Iacono et al. (2008), Mlawer et al. (1997), Fouquart and Bonnel (1980) |
|  |  | Morcrette (2001) |
| turbulence | CBR | Cuxart et al. (2000), Bougeault and Lacarrere (1989) |
| microphysics | ICE3 | Pinty and Jabouille (1998) |
| shallow convection | PMMC09 | Pergaud et al. (2009) |
| deep convection | – |  |
| clouds |  | Bechtold et al. (1995); Pergaud et al. (2009) |
| sedimentation scheme |  | Bouteloup et al. (2011) |
| surface scheme | SURFEX | Masson et al. (2013) |
| LBC scheme | Davies scheme | Davies (1976),Radnóti (1995), Termonia et al. (2012) |

for the semi-Lagrangian advection (Malardel and Ricard, 2015). This scheme allows to use more
conservative horizontal interpolation weights for the variables temperature, wind, specific moisture,
surface pressure, pressure departure and vertical divergence.

The AROME configuration uses a turbulence scheme based on a prognostic equation of turbulent
kinetic energy (TKE), a mass flux shallow convection scheme, a one-moment microphysics prog-
nostic scheme, a detailed surface scheme, and a radiation scheme described below.

The representation of the turbulence is based on a prognostic TKE equation (Cuxart et al., 2000)
combined with a diagnostic mixing length (Bougeault and Lacarrere, 1989). The conservative vari-
ables defined for this TKE scheme are liquid potential temperature, and the total water vapor (addi-
tion of water vapor and cloud water specific contents).

A mass flux scheme (Pergaud et al., 2009) based on the eddy diffusivity mass flux (EDMF) ap-
proach (Soares et al., 2004) is used as parameterization of dry thermals and shallow cumuli. This
scheme uses the same conservative variables as the turbulence scheme. In the boundary layer, the
formulations depend on the buoyancy and on the vertical speed of the updraft, whereas in clouds,
they are computed using a Kain-Fritsch buoyancy sorting (Kain and Fritsch, 1990). Some improve-
ments have been introduced in the latest version of the scheme (more consistent treatment of solid
phase in the updraft, algorithmic corrections).

A statistical cloud scheme is used in AROME (Bechtold et al., 1995; Bougeault, 1982) based
on the computation of the variance of the departure to a local saturation inside the grid box diag-
nosed by the turbulence scheme. The cloud fraction and the cloud condensate content are given by a
combination between a Gaussian and a skewed exponential PDF. The cloud profiles of the shallow
convection are combined with the cloud parameters resulting from the statistical adjustment. Apart
from turbulence and convection, there can be other sources of variance like gravity waves, in par-
ticular with stable conditions when turbulent and convective contributions are too weak to produce





clouds. Following de Rooy et al. (2010), a variance term proportional to the saturation total water specific humidity is added. In this way, the cloud scheme's characteristics are those of a RH-scheme,

where cloud cover is simply a function of the relative humidity.

AROME uses a one-moment microphysics scheme (Pinty and Jabouille, 1998; Lascaux et al., 2006), named ICE3, with five prognostic variables of water condensates (cloud droplets, rain, ice crystals, snow and graupel). ICE3 is a three-class ice parameterization coupled to a Kessler's scheme for the warm processes. Hail is also implemented but not activated in the current version of AROME.

The diameter spectrum of each water species is assumed to follow a generalized Gamma distribution. Power-law relationships are used to link the mass and the terminal fall speed velocity to the particle diameters. More than 25 processes are parameterized in a sequential way inside this scheme. The sensitivity of the scheme to the time step length has been reduced recently through algorithmic changes. A PDF-based sedimentation scheme is used for the numerical efficiency of the micro-

physics computation with relatively long time steps, as described in Bouteloup et al. (2011). In order to investigate the aerosol-cloud interactions, a 2-moment mixed microphysical scheme (Vié et al., 2016) has been developed in Meso-NH and implemented in AROME.

AROME uses the surface modeling platform SURFEX (Masson et al., 2013). Each model grid box is split into four tiles: land, towns, sea, and inland waters (lakes and rivers). The Interactions

between Soil, Biosphere, and Atmosphere (ISBA) parameterization (Noilhan and Planton, 1989) with three vertical layers inside the ground is activated over land tiles. The Town Energy Budget (TEB) scheme used for urban tiles (Masson, 2000) simulates urban microclimate features, such as urban heat islands. Sea tiles use a bulk iterative parameterization, named ECUME (Exchange Coefficients from Unified Multicampaigns Estimates) (Belamari and Pirani, 2007). It is a bulk iterative

parameterization developed in order to obtain an optimized parameterization covering a wide range of atmospheric and oceanic conditions. Concerning inland waters, the classic Charnock (Charnock, 1955) formulation is used. Physiographic data are initialized with the ECOCLIMAP database (Masson et al., 2003) at 1-km resolution. The orography is computed from the GMTED2010 database at 250 m resolution (Carabajal et al., 2011). The FAO HWSD database at 1-km resolution is used

for the fraction of clay and sand in the soil. The HIRLAM parameterization of orography/radiation interactions (Senkova et al., 2007) has been adapted and implemented in the SURFEX version. Orographic shadowing and slopes parameterizations are used operationally to modify solar direct radiative fluxes. One main effect of including shadowing and slopes effects is that the clear-sky sunshine duration is drastically modified in mountainous areas, with values changed from almost constant

to highly varying (sunshine duration can for instance locally reach about zero on grid points with all-day shadow conditions in the French Alps).

AROME uses a simplified version of the European Centre for Medium-Range Weather Forecasts (ECMWF) radiation parameterizations. The shortwave radiation scheme (Fouquart and Bonnel, 1980) uses six spectral bands. Cloud optical properties are derived from Morcrette and Fouquart



(1986) for liquid clouds and Ebert and Curry (1992) for ice clouds. Cloud cover is computed using
       a maximum-random overlap assumption. The effective radius of liquid cloud particles is diagnosed
       from cloud liquid water using the Martin et al. (1994) formulation. Cloud nuclei concentrations are
       assumed to be constant, with one value over land and another over the ocean. The effective radius
       of ice clouds particles is diagnosed from temperature using a revision of the Ou and Liou (1995)
formulation. Long-wave radiation is computed by the Rapid Radiative Transfer Model (RRTM)
       code (Mlawer et al., 1997) using climatological distributions of ozone and aerosols. Ozone monthly
       profiles are given by analytical functions that have been fitted to the U.K. Universities Global Atmo-
       spheric Modelling Programme (UGAMP) climatology (Li and Shine, 1995) with three coefficients
       (Bouteloup and Toth, 2003). The distributions of organic, sulfate, dust like and black carbon, plus
uniformly distributed stratospheric background aerosols, are extracted from the Tegen climatology
       (Tegen et al., 1997). Because of computational constraints, full radiation computations are performed
       once every 15 min.

       The choices of the physics parameterizations of the AROME CMC are summarized in table 2.
       With these settings of the AROME model dynamics and physics parameterizations, the AROME
CMC is capable of capturing in occasionally impressive manner the location, timing and strength of
       intense small scale weather patterns. Fig. 3 is an illustration of a case of onset of severe convective
       precipitation over the French Riviera and the city of Cannes (3 October 2015). For this case, where
       large scale and local effects most likely both are important for triggering the onset of the heavy
       precipitation (more than 100mm in 3h), the model forecast started 15h before the validation time
already provided a very realistic description of the event.

       Météo-France is the main center for the developments of the AROME CMC. The French oper-
       ational implementation, called AROME-France, is the flagship regional forecast system covering
       mainland France and the neighboring regions. The AROME configuration has been first imple-
       mented in operations on 18 December 2008 in Météo France. The current version has a resolution
of 1.3km and 90 vertical levels. The ensemble version and a number of overseas and commercial
       applications are based on a 2.5km resolution, using the same 90 levels, reaching very close to the
       surface.

       The AROME configuration is also, by design, a vehicle for the developments of data assimi-
       lation of high-resolution observational data. Thus the AROME-France initial conditions at model
resolution are provided by an hourly 3D-var cycle for the atmospheric fields and a 3-h Optimal
       Interpolation for the surface fields.

       The performance of the AROME CMC at Météo-France is regularly statistically assessed with
       respect to observations or specific analysis products. The verification encompasses WMO types of
       scores and more focused statistical evaluations as illustrated in Fig. 4. Figs. 4 (a) and (b) show
the frequency bias and the Brier Skill Score for a range of precipitation thresholds for the whole
       year of 2016, respectively. In these two evaluations, the ability of AROME to outperform a rule of





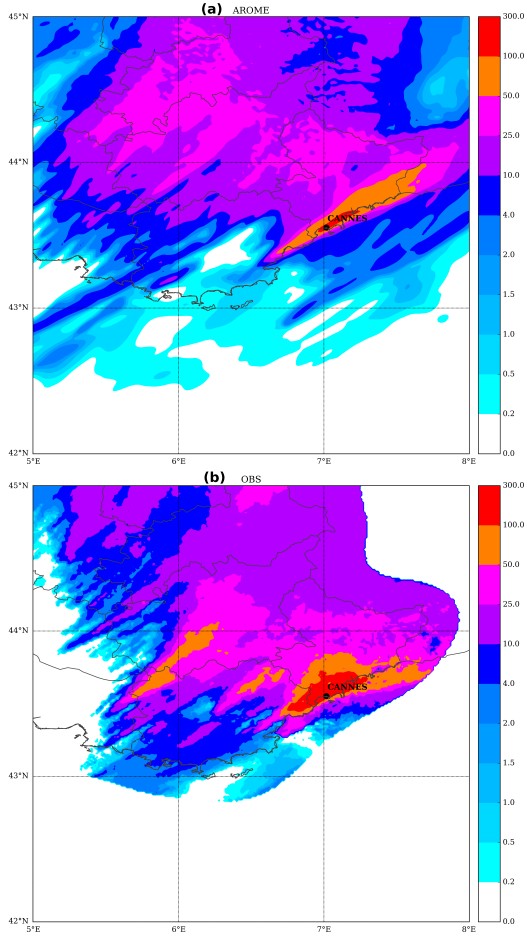

**Figure 3.** Maps of 3h cumulated precipitations between 18 UTC and 21 UTC over the South-East of France, for the case of 3 October 2015. (a) AROME-France forecast starting at 03 UTC, (b) Antilope 3h precipitation analysis taken as proxy to the observation (Laurentin, 2008).

persistence of the forecast is assessed. The reference values, considered as the "truth", are specific analyses of accumulated precipitation obtained from the French ANTILOPE analysis product, which combines radar and rain gauge data (Laurantin, 2008). Ideally, both the frequency bias and the Brier

Skill Score should be one for any threshold (for any event). While obviously the operational AROME system would not exactly reach the theoretical "perfect model" values, the departure from the perfect model results is better appreciated when compared to the results of another modeling system. At Météo-France, AROME results can readily be compared to those of the global ARPEGE system, which are also depicted in Fig. 4. The comparison illustrates that the AROME system significantly





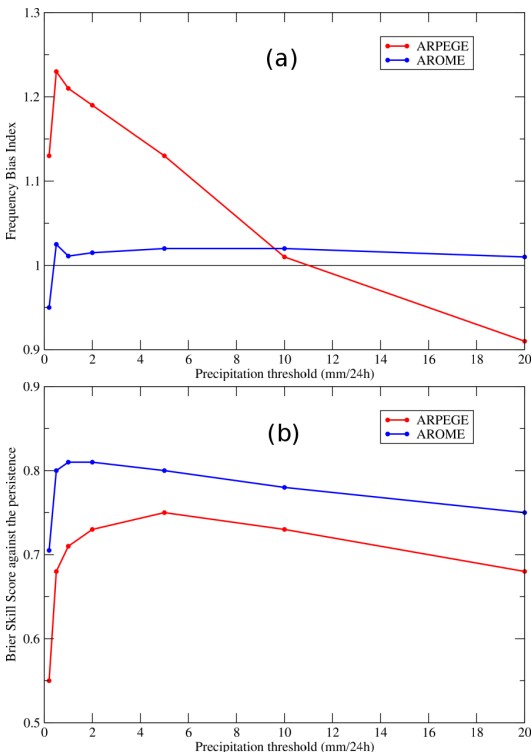

**Figure 4.** Frequency Bias Index (a) and Brier Skill Score (b) against the persistence for a fixed neighborhood of about 50 km (Amodei et al., 2015) computed for 24h accumulated precipitation over France as a function of classes of precipitation (0.2, 0.5, 1, 2, 5, 10, 20 mm/24h). Forecasts are provided by the AROME-France and ARPEGE operational NWP systems at Météo-France, the start time is 00 UTC, the forecast lead time is 30h and the scores are computed for the year 2016. The ANTILOPE precipitation analysis (Laurantin, 2008), combining radar and gauge data, is taken as reference. All data were interpolated on a regular grid of 2.5km.

improves the bias of forecast precipitation amounts as well as the Brier Skill Score for almost all thresholds, with respect to ARPEGE.

### 3.3   The ALARO CMC

The ALARO physics is implemented in the ALADIN System under the same calling routines as the ones for the ALADIN configurations in step 3 of table 1.

The aim of the ALARO configurations of the ALADIN System is to provide a setup that can also be used in intermediate resolutions between the meso-scale and the convection-permitting scales, see Fig. 2. The partners of the ALADIN consortium are running their applications on a variety of computing platforms with different available computing resources. This approach allowed those who



**Table 3.** The ALARO CMC

| parameterization/dynamics | scheme | references |
| --- | --- | --- |
| dynamics for dx > 4km | hydrostatic ARPEGE/ALADIN | Temperton et al. (2001), Radnóti et al. (1995) |
| dynamics for dx < 4km | non-hydrostatic ALADIN | Bénard et al. (2010) |
| radiation | ACRANEB2 | Mašek et al. (2016), Geleyn et al. (2017) |
| turbulence | TOUCANS | Ďurán et al. (2014), Marquet and Geleyn (2013) |
| microphysics | Lopez | Lopez (2002) |
| shallow convection | TOUCANS | Ďurán et al. (2014), Marquet and Geleyn (2013) |
| deep convection | 3MT | Gerard et al. (2009) |
| sedimentation scheme | | Geleyn et al. (2008) |
| orographic gravity wave drag | | Catry et al. (2008) |
| surface scheme | ISBA | Noilhan and Planton (1989) |
| LBC scheme | Davies scheme | Davies (1976),Radnóti (1995), Termonia et al. (2012) |

can not afford to run the model at kilometric resolutions to increase the resolutions in a progressive
way. De Troch et al. (2013) demonstrated the multiscale behavior of ALARO in the statistics of
extreme precipitation in long climate runs.

The basis for this is the application of a multiscale parameterization concept. For moist deep convection, the Modular Multiscale Microphysics and Transport scheme (3MT) has been developed to
overcome problems when convection gets partly resolved at the so-called grey zone model resolutions. The ALARO configuration is built upon this physics parameterizations concept relying on the
governing equations for the moist physics, cast in a flux-form (Catry et al., 2007), a corner stone for
the clean interface between the model physics and dynamics.

From the code point of view, new versions of the schemes are developed by taking utmost care
of the ascending compatibility with the former versions. This allows easier validations, progressive
upgrades and tailoring of the scientific complexity of the local applications. The coding and the
numerical solutions strive for economical use of computing resources and are developed to allow for
the long time steps allowed by the dynamical core. New schemes are also designed to be modular
rather at the level of processes than at the level of full schemes.

The 3MT scheme of moist deep convection develops the idea of separating convective transport
terms and microphysics terms, which indeed happens in Cloud-System-Resolving Models (CSRMs),
Piriou et al. (2007). In this way the moist deep convection problem in the model is tackled by
the separation of dry and moist processes, rather than by a separation of scales, which would be
unnatural.

Moreover, if the precipitation activity terms in cloud budgets models are computed by a microphysics scheme and provided as source terms to the environment, then the system can be closed,




leading to CSRM-type equations that still do not contain detrainment terms. In that case, there is no need to directly rely on the budget equations to close the system.

However, to go from CSRMs to grid-box parameterizations, it is still necessary to cope with sub-grid scale features of unresolved drafts. This is reflected in the schemes of thermodynamic ad-
justment and microphysics. Gerard et al. (2009) used the cloud scheme derived from the proposal of Xu and Randall (1996). Alternatively, the Smith (1990) based formulation can be used. In both schemes the so-called protection of the cloud condensates in the convective updraft part of the grid box is introduced, preventing their re-evaporation by the adjustment. Since the microphysics has to treat condensates of both origin in one go, a geometry of cloud and precipitation has to be included.
Neither cloud, nor the precipitations occupy the whole grid-box, and therefore one has to rely on some assumptions as to their superposition, where the simplest one reflecting physical realism is the maximum-random overlap. The geometry aspect is general; however it is especially fitting to (and rather necessary for) convective cloud and precipitation scenes. Recently, the overlap scheme has been enhanced to the exponential-random one (Hogan and Illingworth, 2000), keeping the orig-
inal maximum-random and fully random cases as limit solutions. The geographical and seasonal variation of decorrelation depth, controlling the transition between maximum-random and random overlaps, was inspired by Oreopoulos et al. (2012). The cloud overlap hypothesis in the microphysical scheme and in the ACRANEB2 radiation are the same for consistency.

Microphysics is at the central position of the 3MT scheme in the organization of the ALARO
CMC physics time step, using a joint input from the adjustment and from the sub-grid-scale updraft condensations. It is modular at the level of processes, which are elaborated based on the work of Lopez (2002). The sedimentation of precipitations is computed statistically (Geleyn et al., 2008) with a variable fall speed of species. Microphysics then provides the input to the sub-grid-scale downdraft computation.

In order to enhance consistency and unification of parameterizations, the strategy employed in ALARO is to go to prognostic, memory keeping schemes (Yano et al., 2016). As an example, in 3MT the convective mesh updraft and downdraft fractions have a prognostic formulation. Similarly, prognostic equations for updraft and downdraft vertical velocities based on the proposal by Gerard and Geleyn (2005) are introduced. The result is a CSRM-type set of equations without any explicit
presence of detrainment terms. In other words, it interacts with the dynamics in the same manner as a CSRM-type of model does.

One can argue that bulk parameterizations should converge in their behavior to the behavior of CSRMs in the cloud-resolving limiting resolutions. If the prognostic equations of the mesh fraction and the updraft-vertical velocity scale properly, then the equations should converge to the equations
of a CSRM. This yields a mechanism to control this convergence and to formulate a scale-aware parameterization of deep convection.





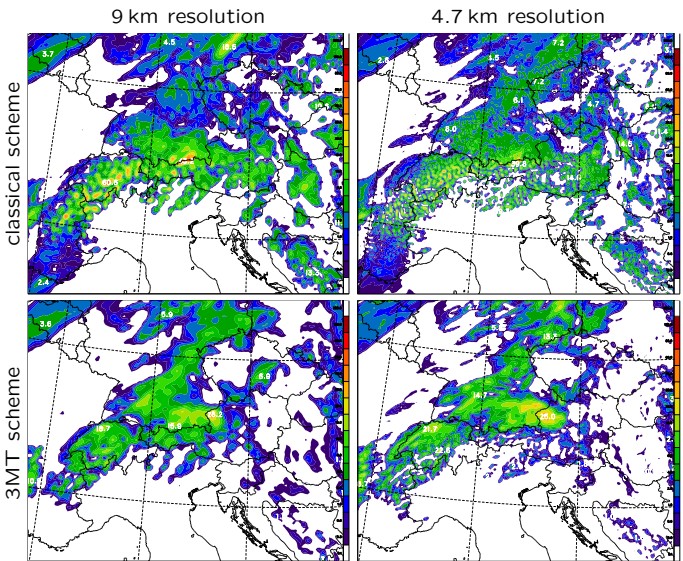

**Figure 5.** Precipitation accumulated between the +12 and +18 hour forecast times starting on 21 June 2006 at 00 UTC given by the ALARO application in CHMI (Prague). The chosen case is a summer convection in Central Europe. The contour levels are 0.1, 0.3, 0.6, 1, 2, 4, 6, 10, 15, 20, 30, 40, 60, 80 and 100 mm/6h.

The 3MT scheme was introduced mid 2008 in a predecessor of the ALARO configuration operations in the application in CHMI (Prague), allowing to increase the resolution to 4,7km, i.e. to enter the grey zone of moist deep convection. It was the world first application of the prognostic

microphysics-transport separation concept in NWP. The multi-scale properties of 3MT are demonstrated in Figure 5, comparing precipitation patterns obtained with a classical steady plume type of moist deep convection scheme (Gerard and Geleyn, 2005) and 3MT at resolutions of 9km and 4,7km.

Recently, good results were found up to a resolution of 1km, when running the so called Grey zone

experiment Cold Air Outbreak case (Field et al., 2016). Further enhancements are currently entering the common library: unsaturated downdraft and complementary sub-grid-scale updraft formulations, which are expected to still improve the convergence of the parameterized moist deep convection to the resolved case (Gerard, 2015; De Meutter et al., 2015).





In the same spirit of separating the precipitating and non-precipitating processes, shallow con-
vection is part of the turbulence scheme TOUCANS (Third Order moments Unified Condensation
and N-dependent Solver). This parameterization of turbulence takes the advantage of recent theo-
retical proposals, such as the revisited Mellor-Yamada system (Mellor, 1973; Mellor and Yamada,
1974, 1982; Cheng et al., 2002; Canuto et al., 2008) , quasi-normal scale elimination (QNSE) the-
ory (Sukoriansky et al., 2005), and energy and flux budget (EFB) theory (Zilitinkevich et al., 2013),
following Ďurán et al. (2014). All of these theories abandon the concept of the critical Richardson
number, beyond which turbulence would cease.

Since TOUCANS can emulate Mellor-Yamada type of stability dependency functions, valid for
all stability conditions, as well as the QNSE and EFB systems; all these models of turbulence are
coded. The ALARO CMC retains the so-called model II of Ďurán et al. (2014). In addition, the
scheme has been extended to a non-local Third Order Moments (TOMs) terms (based on Canuto
et al. (2007)) and to a prognostic equation for moist Total Turbulent Energy (TTE). This concept
makes it possible to better treat the anisotropy of the flow and to account for counter-gradient heat
fluxes.

The introduction of moisture in the turbulence scheme, i.e. accounting for phase changes, leading
to density changes and latent heat release, is based on the recent formulation of moist Brunt-Väisälä
Frequency (BVF) Marquet and Geleyn (2013). The non-precipitating (shallow) convection scheme
of TOUCANS also makes use of this moist BVF, abandoning the older concept of the modified
Richardson number in presence of condensed water. As for the other ALARO schemes, TOUCANS
obeys the governing equations, principles of modularity, memory through prognostic schemes, and
ascending compatibility. Indeed, the older turbulence scheme (Louis, 1979) can be emulated by the
TOUCANS Framework.

An early version of the turbulence scheme TOUCANS having a prognostic TKE treatment (Ge-
leyn et al., 2006), was already introduced in a predecessor of the ALARO configuration and was
put to operations in early 2007 on a horizontal resolution of 9km. It contained a single prognostic
microphysics scheme that jointly handled the inputs from both the thermodynamic adjustment and
the unresolved updraft condensation.

Parameterization of radiative transfer is one of the most expensive computations in NWP models,
therefore a compromise between the cost and accuracy is required. In the case of ALARO the choice
is to keep the cloud-radiation interaction at full spatial and temporal model resolutions, to account
for the fast development and the increased variability of cloudiness that manifest themselves with
the increasing resolutions of the model applications. To achieve this, the ALARO CMC builds on
a broadband approach with single shortwave and single long-wave spectral intervals, where almost
linear scalability of long-wave computations (including scattering) with respect to the number of
vertical levels is obtained via the so-called Net Exchanged Rate (NER) decomposition with bracket-
ing.





Currently, the ALARO CMC offers radiative transfer schemes. The original one denoted ACRANEB is best described in chapter 9.3 of Coiffier (2011), with some components originating from Ritter and Geleyn (1992). Thanks to cheap gaseous transmission calculations based on Padé corrected Malkmus band model, and to statistically fitted bracketing weights, full radiative transfer computations at every model grid-point and time-step are affordable. Somewhat less accurate gaseous transmissions are counterweighted by the full cloud/gas-radiation interaction, ensuring realistic model feedbacks.

The second version called ACRANEB2 (Mašek et al., 2016; Geleyn et al., 2017) was developed with the goal to increase the accuracy of gaseous transmissions, cloud optical properties and the NER technique, while still keeping the full cloud-radiation interaction. Several spectrally unresolved effects had to be parameterized. Cloud optical properties were refitted against modern datasets and the shortwave cloud optical saturation was revised. The computational efficiency of the scheme is ensured by selective intermittency, where rapidly varying cloud optical properties are updated at every model time-step, while slowly varying gaseous transmissions only once per hour. In a shortwave band, gaseous transmissions at every model time-step are updated to the actual sun elevation. In a long-wave band, a two-level intermittency is applied, where the full set of gaseous transmissions needed for the self-calibration of the bracketing weights is calculated only every 3 hours. From the cost versus accuracy point of view, ACRANEB2 is one of the best balanced radiation schemes used in NWP, which makes it fully competitive to the mainstream strategy based on infrequent calls of very accurate but expensive correlated $k$-distribution method. The key point making the selective intermittency affordable is the use of broadband approach, minimizing memory requirements for storing gaseous transmissions.

The choices of the physics parameterizations of the ALARO CMC are summarized in table 3.

The reference versions of the ALARO are maintained in CHMI. Scientifically sound versions are committed during the phasings to the central repository in Météo-France. The ALARO CMCs are created once their model configurations have successfully passed the technical validations mentioned in section 2.1.

Some physics parameterizations can be shared between the two configurations. For instance, the ALARO CMC calls the ISBA surface scheme directly, but it is possible to call the SURFEX scheme from the ALARO configurations. The performance of such an inclusion has been tested by Hamdi et al. (2014) in cycle CY36 of the ALADIN System. Additionally the interfaces to the radiation scheme have been cleaned and the ACRANEB2 radiation scheme (Mašek et al., 2016; Geleyn et al., 2017) of the ALARO configurations can be called from the AROME physics package relying on the common physics-dynamics interface INTFLEX.





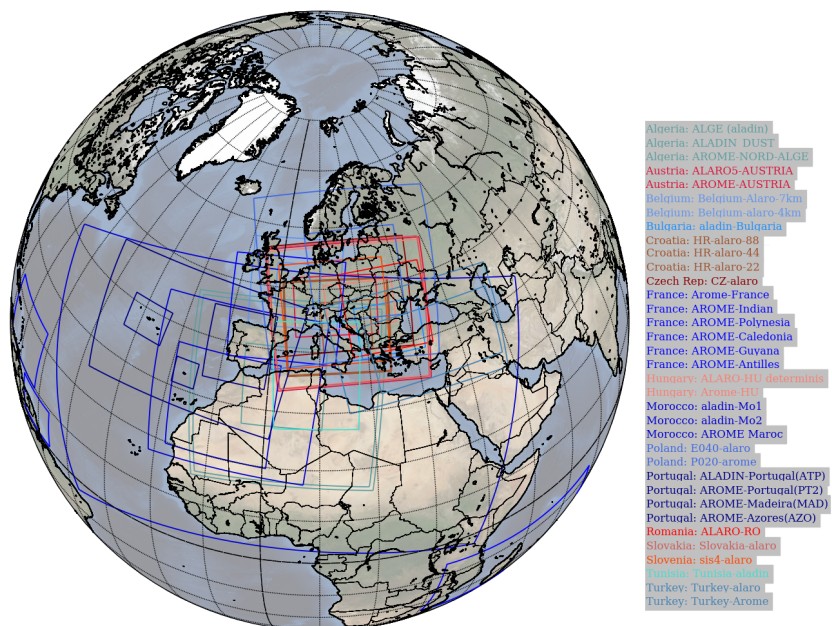

**Figure 6.** The current operational domains of ALADIN System within the ALADIN consortium.

## 4   Operational implementation in the Partner countries

The ALADIN System is run operationally at all 16 Partners' NHMSs on the domains depicted in Fig.
6. The model configurations are coupled the global models ARPEGE or IFS. The lateral-boundary
coupling data is transferred in a timely manner from Météo-France for the ARPEGE model and
from ECMWF for the IFS data. Every Partner adapts the ALADIN System parameters (domain
size, horizontal and vertical resolution, integration length, driving model) according to his needs and

according to his telecommunications and computing capacities. The different operational versions
are named by the partners, referring to the three configurations ALADIN, AROME or ALARO.
Table 4 summarizes the ALADIN System applications in the Partners countries with their main
characteristics.

Typical configurations are run with horizontal resolutions of 1.3 km and 2.5 km for AROME and

about 4-5 km for ALARO. Some Partners run both configurations in a double-nesting setup: for
instance, ALARO (or ALADIN) on a larger domain with a coarser resolution of 4-10 km, driven
either by IFS or ARPEGE global model, and a convection-permitting AROME or ALARO configu-
ration on a smaller domain focusing on the Partner's country and close neighborhoods, that is usually
coupled to the intermediate ALARO (or the ALADIN) model configuration.



**Table 4.** The current configurations of the ALADIN System running in the ALADIN partner countries, with their nationally-used name, horizontal resolution (HRES), domain size, number of vertical levels (NLEV), Version of the ALADIN System, coupling model and the used configuration (ALADIN, ALARO, AROME).

| Partner | Oper. Model | HRES | Domain size | NLEV | Model version | Coupled with | Configuration |
|---|---|---|---|---|---|---|---|
| Algeria | ALADIN-ALGE | 8.00 | 450x450 | 70 | CY40T1 | ARPEGE | ALADIN |
| Algeria | ALADIN-DUST | 14.00 | 250x250 | 70 | CY38T1 | ARPEGE | ALADIN |
| Algeria | AROME-NORD-ALGE | 3.00 | 500x500 | 41 | CY40T1 | ALADIN-ALGE | AROME |
| Austria | ALARO5-AUSTRIA | 4.82 | 540x600 | 60 | CY36T1 | IFS | ALARO |
| Austria | AROME-AUSTRIA | 2.50 | 432x600 | 90 | CY40T1 | IFS | AROME |
| Belgium | Belgium-Alaro-7km | 6.97 | 240x240 | 46 | CY38T1 | ARPEGE | ALARO |
| Belgium | Belgium-alaro-4km | 4.01 | 181x181 | 46 | CY38T1 | ARPEGE | ALARO |
| Bulgaria | aladin-Bulgaria | 7.00 | 144x180 | 70 | CY38T1 | ARPEGE | ALADIN |
| Croatia | HR-alaro-88 | 8.00 | 216x240 | 37 | CY38T1 | IFS | ALARO |
| Croatia | HR-alaro-44 | 4.00 | 432x480 | 73 | CY38T1 | IFS | ALARO |
| Croatia | HR-alaro-22 | 2.00 | 450x450 | 37 | CY36T1 | HR-alaro-88 | ALARO |
| Croatia | HR-alaro-HRDA | 2.00 | 450x450 | 15 | CY38T1 | HR-alaro-88 | ALARO |
| Czech Rep | CZ-alaro | 4.71 | 432x540 | 87 | CY38T1 | ARPEGE | ALARO |
| France | Arome-France | 1.30 | 1440x1536 | 90 | CY41T1 | ARPEGE | AROME |
| France | AROME-Indean Ocean | 2.50 | 900x1600 | 90 | CY41T1 | IFS | AROME |
| France | AROME-Polynesia | 2.50 | 600x600 | 90 | CY41T1 | IFS | AROME |
| France | AROME-Caledonia | 2.50 | 600x600 | 90 | CY41T1 | IFS | AROME |
| France | AROME-Guyana | 2.50 | 384x500 | 90 | CY41T1 | IFS | AROME |
| France | AROME-Caribbean | 2.50 | 576x720 | 90 | CY41T1 | IFS | AROME |
| Hungary | ALARO-HU determinis | 7.96 | 320x360 | 49 | CY38T1 | IFS | ALARO |
| Hungary | Arome-HU | 2.50 | 320x500 | 60 | CY38T1 | IFS | AROME |
| Morocco | Aladin-NORAF | 18.00 | 324x540 | 70 | CY41T1 | ARPEGE | ALADIN |
| Morocco | ALADIN Maroc | 7.50 | 400x400 | 70 | CY41T1 | ARPEGE | ALADIN |
| Morocco | ALADIN Ma 3DVar | 10.00 | 320X320 | 60 | CY36T1 | ARPEGE | AROME |
| Morocco | AROME Maroc | 2.50 | 800x800 | 60 | CY41T1 | ALADIN Ma 3DVar | AROME |
| Poland | E040-alaro | 4.00 | 800x800 | 60 | CY40T1 | ARPEGE | ALARO |
| Poland | P020-arome | 2.04 | 810x810 | 60 | CY40T1 | E040-alaro | AROME |
| Portugal | ALADIN-Portugal(ATP) | 9.00 | 288x450 | 46 | CY38T1 | ARPEGE | ALADIN |
| Portugal | AROME-Portugal(PT2) | 2.50 | 540x480 | 46 | CY38T1 | ARPEGE | AROME |
| Portugal | AROME-Madeira(MAD) | 2.50 | 200x192 | 46 | CY38T1 | ARPEGE | AROME |
| Portugal | AROME-Azores(AZO) | 2.50 | 270x360 | 46 | CY38T1 | ARPEGE | AROME |
| Romania | ALARO-RO | 6.50 | 240x240 | 60 | CY40T1 | ARPEGE | ALARO |
| Slovakia | Slovakia-alaro | 4.50 | 576x625 | 63 | CY36T1 | ARPEGE | ALARO |
| Slovenia | sis4-alaro | 4.40 | 432x432 | 87 | CY38T1 | IFS | ALARO |
| Tunisia | Tunisia-ALADIN | 7.50 | 216x270 | 70 | CY38T1 | ARPEGE | ALADIN |
| Turkey | Turkey-alaro | 4.50 | 450x720 | 60 | CY38T1 | ARPEGE | ALARO |
| Turkey | Turkey-Arome | 2.50 | 512x1000 | 60 | CY38T1 | ARPEGE | AROME |

The installation and upgrade of the ALADIN System is performed by the Partners individually, thanks to the high level of expertise gathered in each NHMS during the past course of the ALADIN project. Dedicated and coordinated efforts are made to support the installations of the newest cycle at Partners' NHMS in order bring to them at a state-of-the-art level, allowing to implement the newest





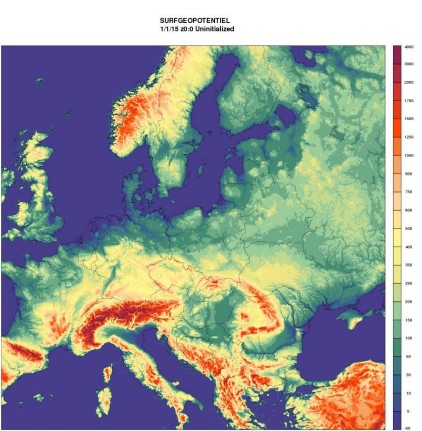

**Figure 7.** The current domain of the Polish operational CY40T1 ALARO-CMC.

research and development achievements. An example is the upgrade of the ALARO CMC where the
corresponding source code was provided by CHMI on their actual cycle (CY38T1) and phased to
the cycle just being prepared (CY40T1) to allow all Partners to benefit from the new developments.
This support comprises the collection and redistribution of information about known problems and
their fixes.

### 4.1 Implementation of the CMC's

By using the canonical configurations the ALADIN partners can be sure that they are running a
configuration with physically consistent choices. Currently the ALADIN consortium is installing
cycles CY40T1 and CY41T1 that are described in section 3 in its operational applications. As an
illustration, the CY40T1 ALARO-CMC (ALARO-1) has been ported to operations in Poland, see
table 4. It is running on the domain shown in Fig. 7. It is running with a 16 points wide coupling
zone, a 3-h coupling to ARPEGE CY40T1. There are 4 operational forecasts per day at 00, 06, 12,
18 UTC with respective forecast ranges of 66, 66, 66, 60 hours. The model has been validated by the
ALADIN team at the Institute of Meteorology and Water Management (IMWM). They have shown
that it improves on the previous version of the ALARO CMC, called ALARO-0. Some scores are
shown in Fig. 8. As can be seen from table 4, not all Partners have made the switch to the latest CMC
at the time of writing. Although it is strongly encouraged to follow the new cycles, some Partners
may still use older Versions in some cases.

In terms of local implementation, the operational ALADIN System configurations mostly focus
on the need to provide a state-of-the-art forecasting system with convective scale resolution. The
goal is to provide forecasters, other production departments in ALADIN national weather services,





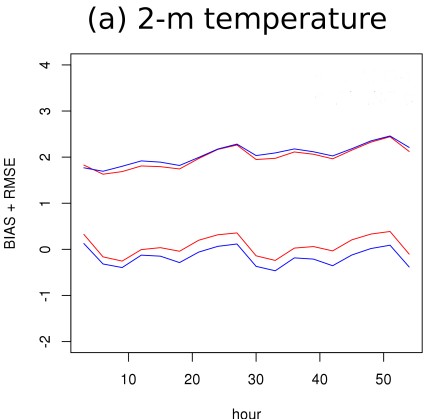

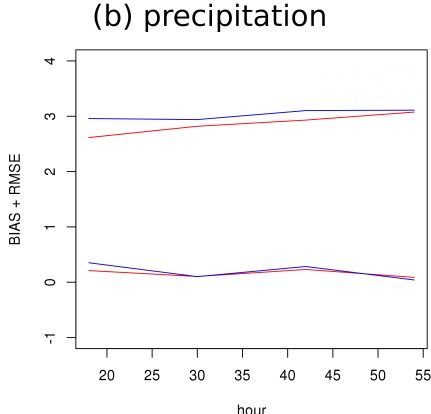

**Figure 8.** The bias and the root-mean-square error (RMSE) for the Polish ALARO version of CY38T1 (indicated as ALARO-0 in blue) and the ALARO CMC of CY40T1 (indicated as ALARO-1 in red) configurations for (a) 2-m temperature (K) and (b) 12-h accumulated precipitation amounts (mm). The verification is done for 2013 using 60 synoptic stations in Poland.





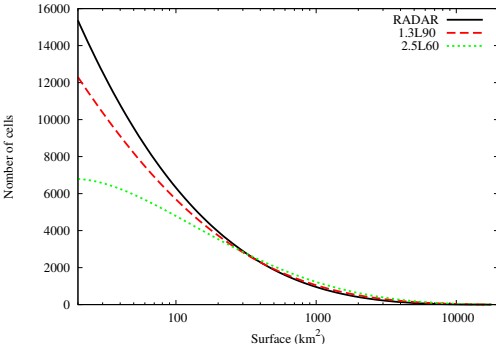

**Figure 9.** Distribution of the number of convective cells against their size represented by an estimate of the cloudy area, as derived from the data of the French radar network (40 dBz reflectivity detection level, solid black curve), from the 1.3km/90 level AROME version (dashed red curve) and from the 2.5km/60level AROME (dotted green curve). The statistics have been aggregated over 48 convective days of 2012. Adapted from Brousseau et al. (2016).

and eventually stakeholders and users of various type, an added value forecast of severe weather outbreaks, very local weather patterns and a variety of meteorological output fields and products. A typical example of severe weather of concern is heavy precipitation and strong convection, with their possible associated features like severe wind gusts, heavy hail or flooding.

The progressive increase of resolution led to more realistic forecasts of convective systems. As an 715 example, Fig. 9 displays the number of convective cells as a function of their size, represented by the cloud-covered area, derived respectively from the observations of the French radar network, the 2.5-km version of AROME-France, and the newer 1.3-km version (Fig. 9 is adapted from Brousseau et al. (2016)). The new version of AROME provides a more realistic distribution of cell size, with both a larger amount of small cells, as suggested by the radar data, and a slight decrease of the 720 number of large ones. Brousseau et al. (2016) also reported an improved timing of the diurnal cycle of convective activity, improved scores of accumulated rainfall thresholds or wind gusts.

The new Versions of the ALADIN System are also verified for specific past cases that are of primary interest, demonstrating added value of the high-resolution forecasts with respect to the global model or with respect to the previous versions. Fig. 10 shows an example of a warning of 725 the AROME configuration AROME-Aut[11] running in Zentralanstalt für Meteorologie und Geodynamik (ZAMG) (see table 4). It is the June 1st 2016 forecast of a flash flood event that took place at the border region between Austria and Germany. Fig. 10a shows the 24 hour accumulated INCA

---

[11] This version uses a combined 3DVAR for the atmosphere and an optimal-interpolation (OI) for the surface to create the initial conditions. The lateral boundary conditions with hourly resolution are created from the IFS high-resolution (HRES) model.





**Table 5.** 24 hour accumulated area mean and area max values for the region (longitude /latitude: 12.75 - 13.5 / 47.65 - 48.45) for INCA, AROME-aut and IFS-HRES.

|  | Area mean [mm/24h] | Area max [mm/24h] |
|---|---|---|
| INCA analysis | 58.0 | 141.6 |
| AROME-Aut | 41.5 | 137.5 |
| IFS HRES | 26.6 | 41.3 |

precipitation analysis (combination of rain gauge and radar data, see Haiden et al. (2011)) for Austria and the surrounding regions. It can be seen that the observed values exceeded 100mm in 24 hours.

However, the intensity of the flooding observed in this region and the river gauge measurements indicate that local maxima of precipitation must have been significantly higher than 100mm/24h up to even 200mm/24hours. Figs, 10b and 10c represent the corresponding precipitation forecast for AROME-Aut and IFS HRES. One can see that the localization of the strongest activity is captured well in both models, AROME and IFS, but the overall amplitude is much better simulated

by AROME-Aut. This is confirmed when considering the area mean and area max values of INCA, AROME-Aut and IFS HRES in table 5. The area values shown are computed for a rectangular region indicated by a yellow square in Fig. 10.

Efforts are made to steadily increase the resolutions of the applications. For instance, the operational viability of the CY40T1 ALARO CMC is tested at km-scale resolution over Belgium by

the Royal Meteorological Institute of Belgium (RMI), as represented the lower part of the diagram in Fig. 2. It is a regular 1.3 km grid on a Lambert projection, with its center at (50.57 N, 4.55 E), with 588 physical gridpoints in the East-West and North-South directions, and with 87 vertical layers. This ALARO CMC run at km-scale was evaluated for a severe convective storm of 18 August 2011 causing casualties at the Pukkelpop music festival in Belgium, see De Meutter et al. (2015).

Fig. 11 presents the accumulated precipitation between +06h and +30h forecast ranges simulated by ALARO and observed with the Radar of Wideumont of the Royal Meteorological Institute, Belgium (Delobbe and Holleman, 2006). The red dot presents the location of the Pukkelpop music festival. The newer version of ALARO reproduces the location and the amount of precipitation for this storm better than the current operational version that is run at 4-km resolution.

It should be mentioned also that CMCs of the ALADIN System are being used with data assimilation, with ensemble prediction systems (EPS) and with rapid update cycles for nowcasting purposes. For instance, the AROME CMC is operationally implemented in Météo-France's nowcasting system (Auger et al., 2015) and in five Overseas 2.5km versions (Soutwestern Indian Ocean, Caribbean, French Guyana, Polynesia and New Caledonia). A 12-member ensemble prediction system using

AROME at 2.5km resolution, named PEARO (Bouttier et al., 2016; Raynaud and Bouttier, 2016), is also daily running on Météo-France's computing system. A detailed description of the activities re-



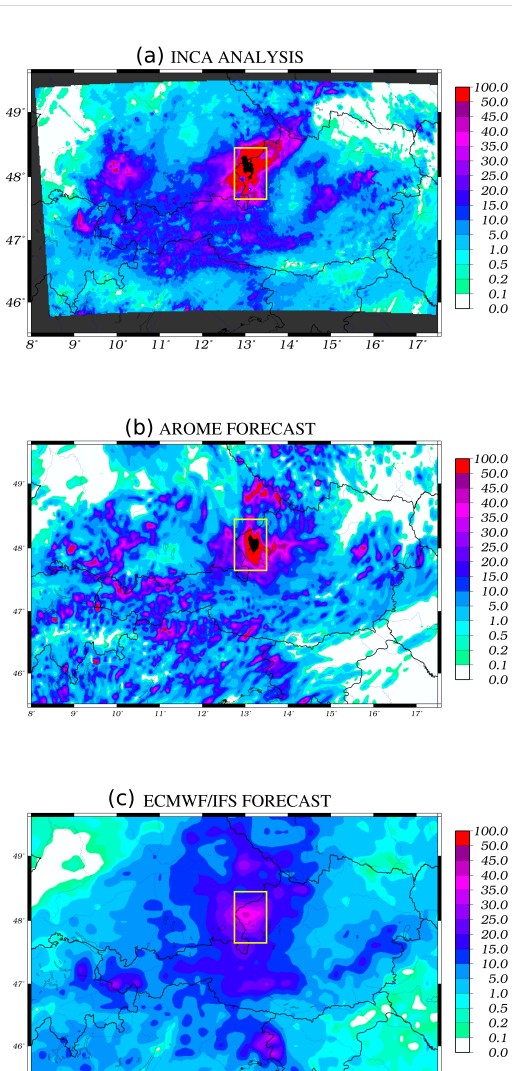

**Figure 10.** (a) INCA precipitation analysis for the 24 hour period of 20160531 12 UTC - 20160601 12 UTC, (b) AROME-Aut 24 hour accumulated precipitation forecast for the period of 20160531 12 UTC - 20160601 12 UTC (Initialization time: 20160531 12 UTC), and (c) IFS-HRES 24 hour accumulated precipitation forecast for the period of 20160531 12 UTC - 20160601 12 UTC (Initialization time: 20160531 12 UTC).

garding data assimilation, EPS and nowcasting within the ALADIN consortium is outside the scope of this paper.





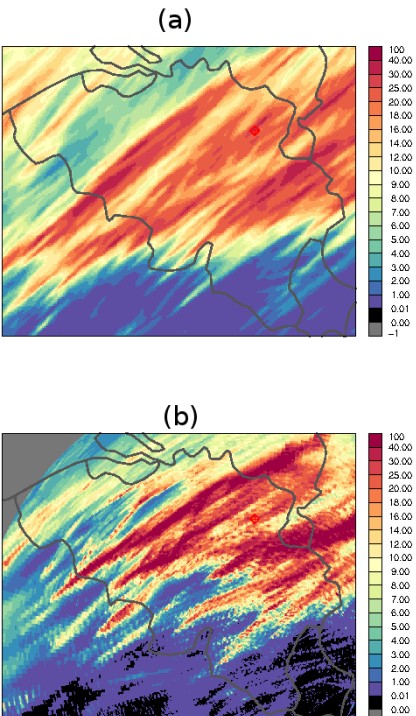

**Figure 11.** The accumulated precipitation between +06h and +30h forecast range (a) simulated by ALARO and (b) observed with the Radar of Wideumont, Belgium.

### 4.2 Tailor-made configurations

Configurations of ALADIN System are used by the partners of the consortium for scientific studies. In many cases, the partners rely on their own expertise to adapt the Versions of the ALADIN System to develop tailor-made tools for their national needs.

As an illustration, the configurations of the ALADIN System of Croatia (shown in table 4) have been used for dynamical adaptation of the wind field to 2-km resolution since 2000, see Ivatek-765 Šahdan and Tudor (2004).

ALARO-HRDA has had a large success in forecasting spatial and temporal variability of local windstorm Bura (Grisogono and Belušić (2009)). The high resolution wind field forecast has been an essential ingredient issuing warnings for hazardous weather and safety of traffic at sea and on land. ALARO-HRDA was used to create a wind atlas of Croatia by downscaling the ECMWF ERA40 770 reanalysis data (Uppala et al., 2005) through ALARO-88 as an intermediate step.




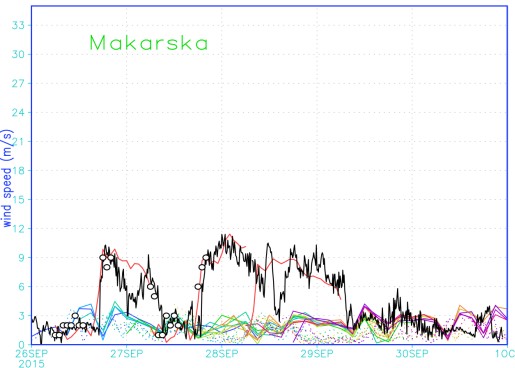

**Figure 12.** Wind speed in Makarska (lon 17.02, lat 43.28) from 00 UTC 26 September 2015 until 00 UTC
1 October 2015, measured by the local automatic station with 10-minute intervals (black), the local synoptic
station (black circles), and forecasts: HR-alaro-22 (red), HR-alaro-88 (full lines) and HR-alaro-HRDA (dashed)
are plotted in rainbow sequence depending on the analysis time (blue for the run starting at 00 UTC 26 Sep
2015, light blue for 06 UTC the same day etc.).

There are episodes of severe Bura associated to local dynamical phenomena that require high
resolution forecasts using non-hydrostatic dynamics and complete ALARO physics package (Tudor
and Ivatek-Šahdan, 2010). The ALARO configuration has been adapted by the Meteorological and
Hydrological Service to run at a resolution of 2 km, the so-called HR-alaro-22 (indicated in table
4). It is in the operational suite since July 2011. The wind field forecast is improved (Figure 12) for
local short burst events. This ALARO configuration of the ALADIN System uses the ALADIN NH
dynamics, the ALARO physics package, the SSDFI for initialization and is coupled to the global
model with a 1-h coupling-update frequency.

Configurations of the ALADIN System are still used for applications where meso-scale applica-
tions are required, for instance, there are regional-climate model versions of ALADIN and ALARO,
as mentioned above. An ALADIN configuration is used by the UERRA project (FP7 project) to pro-
vide an atmospheric European re-analysis (3Dvar) at 11km over Europe for the period 1961-2015
[12].

## 5   Discussion and outlook

The aim of this paper was to describe the current state of the forecast model configurations of the
ALADIN System and review the rationale behind the scientific options made in the past develop-
ments of the ALADIN System. Given the increase of choices in the model configurations, the AL-
ADIN consortium introduced the notion of Canonical Model Configurations. These are privileged,

---

[12] see its project web site www.uerra.eu



physically-consistent configurations that are intensively validated and for which support from the
consortium is provided to implement them as operational applications in the ALADIN Partner coun-
tries. The status of the current two CMCs AROME and ALARO was described and a status report on
their validation and implementation in the ALADIN Partner's NWP applications was given. While
doing so this paper clarified the meaning of the acronyms used within the ALADIN consortium.

The scope of the present paper was limited to the forecast model configurations, excluding data
assimilation, EPS perturbation methods, post-processing software, scripting systems and so forth,
but relevant references to these systems were given throughout the paper without aiming to be ex-
haustive.

The ALADIN consortium provides a platform for the ALADIN members for organizing optional[13]
activities related to numerical weather prediction. This can be done by individual members or in more
intense optional multilateral collaborations. The applications range from nowcasting tools, specific
academic case studies, to past and future climate simulations. Long model runs are used for creating
high-resolution wind-climate atlases.

Codes developed within the context of the cooperation agreement with the HIRLAM consortium,
have been colloquially called HARMONIE[14] in the past. Recently Bengtsson et al. (2017) clari-
fied the meaning of the acronym HARMONIE. HIRLAM adapted the AROME CMC to create its
HIRLAM reference configuration and this is called the HARMONIIE-AROME configuration. It has
been decided to limit the meaning of the acronym HARMONIE to this configuration only. In other
words, the acronym HARMONIE does not cover to the configurations of the ALADIN System. The
model configurations used in Termonia et al. (2012) were configurations of the ALADIN System. Of
course, the schemes presented in that paper can also be applied in the HARMONIE-AROME con-
figuration but they should not be understood as being restricted solely to the HARMONIE-AROME
configuration.

The shared codes are undergoing a number of code modernizations driven by the strong will to
keep them fit both for optimal use of upcoming high-performance computing architectures and for
further scientific and meteorological evolutions. This is a significant investment, performed together
with ECMWF. Its involves the use of object-oriented software layers to provide a further abstraction
level in data assimilation on the one hand, and in compute grids on the other hand, accompanied
by disentangling and modularization, optimization and portability issues (including reliability on
massively parallel HPC). Extra work on the development of scripts for data assimilation is planned.
There are no short-term reasons to abandon the spectral numerical techniques of the dynamical core
of the ALADIN System as long as the inherent scalability weakness is more than balanced by the
advantage of being able to run with large Courant numbers. Nonetheless, the ALADIN consortium
carries out research on scalability and efficiency issues including the study of local discretization

---

[13] Optional activities mean that the ALADIN consortium does not per se, today, provides coordination for these activities
among its members, but facilitates them through the management and the delivery of the codes of the ALADIN System.

[14] HARMONIE stands for HIRLAM ALADIN Research on Meso-scale Operational NWP in Euromed





methods with research studies ranging from adapting the semi-implicit problem formulation and
825 solution to try and keep the large Courant number time-stepping, to being able to solve the same
equations using a HEVI (horizontally explicit, vertically implicit) scheme, the latter being a kind of
fall-back solution.

*Code Availability.* The ALADIN Codes, along with all their related intellectual property rights, are owned by
the Members of the ALADIN consortium and are shared with the Members of the HIRLAM consortium in
830 the frame of a cooperation agreement. This agreement allows each Member of either consortium to license the
shared ALADIN-HIRLAM codes to academic institutions of their home country for non-commercial research.

*Obtaining the ALADIN System codes.* Access to the codes of the ALADIN System can be obtained by con-
tacting one of the Member institutes mentioned in the introduction of this paper or by sending a request to
patricia.pottier@meteo.fr and will be subject to signing a standardized ALADIN-HIRLAM License agreement.

835 *Acknowledgements.* The activities of the ALADIN consortium started in 1991 after an initiative taken by
Météo-France. The current system is the result of the contributions of many experts from the ALADIN, the
ARPEGE and the IFS communities. The merits of the authors in the developments of the ALADIN System are
small compared to this. The present paper is meant to give a status review of the current system according to
our best efforts. While the list of contributors is too long to be acknowledged here, we point out the unique
840 contributions of the late Jean-François Geleyn. He was the driving force behind the creation of the ALADIN
consortium and he was the leading scientist of the developments of the ALADIN System. His vision further
enabled the training of many young scientists throughout Europe and North-Africa to state-of-art numerical
weather prediction. When he passed away in 2015 the consortium lost an exceptional mind. We dedicate this
paper to his memory.





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
