# Peer review of "The ALADIN System and its Canonical Model Configurations AROME CY41T1 and ALARO CY40T1"

_Geoscientific Model Development, 2017_

## Referee Comment (RC1) · Anonymous Referee #1 · 15 Aug 2017

This paper aims to set a general terminology related to the limited area model (LAM) development, maintenance and scientific evaluation within the complex environment of the ALADIN international consortium. Two basic configurations are explained in detail serving as a reference canonical configuration. In addition they are also nicely illustrating the nearly infinite flexibility allowed by the common source code. This specific LAM version of the whole software is also influenced by the global model development at ECMWF and Meteo-France, being mostly based on the same code. This with the fact the ALADIN consortium is represented by 16 member states while mutually interfering with the other partners from HIRLAM consortium (and ECMWF) makes the work-flow and terminology around this software very challenging and bit cryptic for people not

directly involved.

The attempt to make this nomenclature more transparent plus offering quite detailed description of the two basic canonical configuration is naturally making this paper worth to be published. It will be certainly not just for a benefit to the ALADIN consortium members but can serve also as a very useful reference for the modellers from other communities. The usefulness of such summary paper could be also documented by the recent publication of Bengtsson et all documenting the HARMONIE system being a specific sub-group of the ALADIN software.

It is of no surprise that such an attempt is logically a result of several authors contribution. To serve however for the intended purpose the text needs some additional editorial work. Some parts are not very consistent. Some are duplicated and for some parts one just wonder why they are incorporated. The paper is relatively long. It would be thus of great benefit when the readability and its self consistence is improved. Apart from this the paper is certainly worth to be published. I have, however, a number of issues which I hope the authors can address.

General comments:

1/ The text on pages 20-21 (especially lines 544-574) is quite cryptic and inconsistent with the rest of the paper. Here rather than documenting the particular physics package it looks like some ideas or guidelines are presented. They are however hardly justified by some published evidence or experience from other NWP centers. The ACRANEB2 radiation is used without a word of introduction there. The explanation is jumping from microphysics to convection. The presented text is not self-explanatory: What is meant for example "modularity at the level of processes"? I suppose every microphysics scheme is in a way modular by describing separately process by process. Some extensive revision and shortening of this text resulting ideally in description on what has been actually implemented would be highly desirable. (Without too much arguing WHY: This should have been published already elsewhere.)

[Figure]

2/ The text has the ambition to serve as an reference (for the ALADIN and also external community). To help this purpose it would greatly improve the navigation when the schemes described in section 2.2 are highlighted (in bold or italic) near the area they are described. When they are mentioned later on in the text a general reader would easily search for them even working with printed text. (For example by highlighting the SSDFI at L276 you will improve the readability of the text at L777.) This could be seen as a poor man's glossary.

3/ Some part of the text looks like fillers. A reader may not see a clear reason for incorporating those into the paper. For example Figure 7 brings no specific extra information. It would make perhaps more sense if there is a comparison of the new and old Polish domains. But the same could be summarized easily without a figure. Especially when there is already Figure 6 illustrating roughly the same. Instead of complicated and sometimes very case specific results from newer model versions one would expect to find some highlight parameters of scores from global models (those used for boundary conditions), reference LAM version and new LAM version. This is clearly missing.

There are also plenty of references having no relevance to the paper. As the data assimilation is out of the scope of this paper, a reader may ask why there are so many papers referred to this subject? Some other references are definitely not the most appropriate to the presented subject. Please be honest and provide only relevant references to the presented text.

4/ On page 5-6 you describe a procedure of a new model version assembling. It is not clear however how the evolution of the global model code is interfering with this. How the decision about what is implemented at the level of the LAM code is taken? Or does it mean everything developed for global model automatically propagates to the LAM? What about some specific global model issues which are not relevant for the LAM community (like specific treatment of poles)? Do you have some general guidelines or those are solved on case to case basis? Is there some experience with the opposite

direction code propagation, like a code developed initially for the LAM model has been made available for the global model too? This perhaps rather particular question tries to reveal a bit more about this rather unique duality that the same code is used for LAM and global model communities.

Specific comments & Technical corrections:

1/ Purpose of the paper is given twice: p1/10 and once again p4/90-102. Could it be perhaps unified and reduced to just one list?

2/ Duplicated text describing the paper limitation: See p1/L15-6 and once again p4/L103-4. In this latter case the repetitive text brings just references to additional papers having no relevance to the described CMCs.

3/ p5/L132 text mentions a five-step process defining the scientific developments of the ALADIN System. Apparently there is no such description given in the paper (or it is well hidden). What this "five-step process" should be representing then? When it is given somewhere in the text please make it more explicit to be obvious without an extensive search.

4/ p7/L182-5 Could you explain what is the driving force for you to insist on long time steps? Is it the computational efficiency? Or do you have some specific scientific reason for it? (The computational efficiency doesn't need to be necessarily always justified by long time steps.) This claim feels bit like a dogma. But it is not clear for a reader why this is so important here.

5/ p7/L195 Is the best reference for the SI scheme really the given papers from Caluwaerts et al?

6/ p8/L215 Could you bit develop on this claim relating the 3km threshold and "important" role for the non-hydrostatic dynamics? First, it is not clear how this threshold is defined in terms of model: Are you referring here the grid point distance of the computational mesh, the shortest wave represented by the model or even a size of the

smallest fully resolved feature of the model? Second, please specify the "important role". Could you perhaps give some reference to clarify this claim? To the reviewer's knowledge there is no clear agreement on it. One can perhaps find some effects not simulated with hydrostatic dynamics at those scales. But this still doesn't justify the necessity to use non-hydrostatic dynamics there. One can argue that the non-hydrostatic schemes are only essential when it comes to the simulation of the convective effects. Here we are however referring scales bellow ∼100m of grid resolution. Finally, the role of "details of the used numerical scheme" is also not very clear here. Do you mean the true resolution given by the particular numerical scheme? Or something else is meant? To conclude: this sentence sounds like referring some common truth. If there is such an evidence, please provide some reference. Alternatively please make this statement less controversial.

7/ p8/L220 For VFE there is more fundamental reference of Untch and Hortal to be used rather than the one given in the text.

8/ p9/L256 When you give the diffusion order, you should also specify the resolution (and/or) truncation. Does it mean all presented configurations from Table 4 are running with this 4th order horizontal diffusion (including 18 km Aladin-NORAF and 1.3 km Arome France)?

9/ p11/L291-3002 It is nowhere specified how wide the relaxation area is. From the text at p27 it is apparent the number of coupling zone points is varying. How the given values of parameters p (L301) are modified with respect to the changing z? It is quite evident the optimal value of p must be related to the number of points in the coupling zone and model resolution. Can it be precise here?

10/ Several places like p12/L354 and p13/L363,364 are using term "dual parallelization". This is not at all very common term. Presumably it is meant mixed or hybrid MPI/OpenMPI parallelization? If so please change it to hybrid parallelization which is more commonly used name.

11/ p16/L448 The sensitivity of the scheme to the time step length has been... changes. This sentence brings no information to a general reader. Please either provide some details or drop it.

12/ p17/L487 This 15 minutes intermittency is used at every Arome configuration? Your example is given with Arome 1.3 km and 50s time-step. But there are some 2.5 km and 90s Arome configurations in the Table 4. Does it mean the 15 minutes remains fixed regardless the actual time-step length?

13/ p14-p19 (Arome CMC) At scales of 1.3km is certainly not negligible a contribution of horizontal mixing/turbulence. Please give some details about your representation of those highly non-liner horizontal effects.

14/ p20/L546-8 Separation of scales is not unnatural. I believe it is meant rather arbitrary. The separation of processes to dry and moist is equally unnatural/arbitrary, by the way. The text is not correct. There can't be such clear separation. This just says the microphysics is called twice in this case.

15/ p22/23 Could you specify the closure used for the turbulence scheme? Is it closed by a mixing length? And if so, which one?

16/ p23/L622-626 Rather strange text with a link to turbulence but then mentioning microphysics. What is the relevance of it? Does the microphysics influences the turbulence?

17/ p24/L654-5 This is rather strong claim. Could you perhaps give some reference or bring some more evidence supporting it?

18/ p22/24 Can you give some description for the microphysics and gravity wave drag parameterization? A reader may wonder what makes those two schemes so unattractive that the only information about them can be found in the table 3.

19/ The paper of Lopez(2002) being referred as the microphysics description is introducing only three prognostic variables: water vapour, cloud condensates and falling

precipitation. Is this really the case for the presently used microphysics? If not could you explain the choice of prognostic variables related to the microphysics in ALARO CMC?

20/ p25/L671: Missing "with" or "to"?

21/ p26/Table 4. Please specify the date of validity. The actual state could be evolving.

22/ p28/Fig 8 Are the curves based on annual verification of the two models? If so it is truly impressive, but better to say it more explicitly. In the other case please specify the verification period. It would be also useful to add the zero horizontal line (especially to the upper panel) in order to help the results interpretation.

23/ p32/Fig 11 The red dot is nearly invisible (especially when printed). Please use some better way to highlight it. This figure demonstrates the superiority of the newer version of ALARO over the operational one. Could you then add the operational results to illustrate it graphically?

24/ p33/Fig 12 Could you please zoom the figure to its lower third? It is really difficult to follow the presented the multiple lines of HR-alaro-88 and HR-alaro-HRDA.

---

## Referee Comment (RC2) · P. Unden (Referee) · 18 Aug 2017

The paper is the first full description of the ALADIN system and it is very valuable to have all its components with references to underlying research and papers for all the modelling components, in one paper.

The assembly of modelling components and their configurations into these CMCs is novel and clearly described. Most of the components have been developed during the lapse of the ALADIN history but several of them are from recent years and documented through the references but still described and explained in some detail. In addition there are some new developments mentioned in the paper, but they are more options and

not yet included as defaults in the CMCs.

There are a number of results to demonstrate aspects of the model, some already published and some that appear to be new. They are mainly there to illustrate but are not meant to document the construction of the model (which can be found in references and which is far too much to be in an overview paper like this).

The title, abstract and the presentation are all very clear and relevant. The text, language and flow through the paper are all of high quality and well written and easily readable.

The content and number of references are sufficient and there are only some aspects that need to be clarified and added to be able to compare the CMCs. In particular, ALADIN baseline CMC is not suffiently described to be able to compare with the other ones. See the detailed comments below. It is important since also the global ARPEGE often used as a host model, has the same physics package as I understand.

I would support a publication with minor revisions based on the overall scientific significance of all the components even if the paper does not attempt to present any one particular new idea of method that e.g. can be copied or reproduced. The presentation quality is very high.

Detailed comments:

1. typo: line 273 p 10: In operation . . .. should be operational

2. line 240 p 9 : . . .. more conservative semi-Lagrangian . . ... : please make the link to the same but a bit longer explanation of this scheme around line 413. Perhaps also here refer to it as COMAD to make it consistent.

3. line 420 p 11 : Please make the comparison with the TKE scheme in AL-ADIN/ARPEGE on line 391. From the text it appears to be the same scheme albeit with some different variables but it is relevant here to state what is shared and what the differences are between the TKE schemes, or indeed if they are or could be the same

or share the same code.

4. line 439, p 16: In this way …. of a RH-scheme …. : I don't understand this at all. The earlier sentences all give the message that the scheme is everything but a RH scheme! Which of the "ways" just mentioned makes it a RH scheme? Please qualify and explain or change if it is an error.

5. line 450 p 16 2-moment scheme …. implemented …. : please add something like not activated since on 441 you describe the current one moment scheme, confusing for the non-initiated.

6. Line 473 p 16: Again, please compare with ALADIN radiation on line 388. There are many common components in the basic scheme it seems.

7. typo line 499 : Météo … - missing

8. Before Table 2. There should be a Table for the ALADIN baseline CMC as well – to be able to compare AROME and ALARO!

9. Figure 8. Please state if it is for the whole year of 2013 or which period.

---

## Author Response (AR1)

**The ALADIN System and its Canonical Model Configurations AROME CY41T1 and ALARO CY40T1**

**Reply to the comments of reviewer 1**

Indeed, this paper documents a NWP system that is the result of more than 25 years of R&D activities, and it is quite a challenge to document this. Reviewer 1 is right that this paper became the result of many contributions from many authors. All of his/her comments are relevant and to the point. We took the opportunity to implement them in a revised manuscript and we believe it improved a lot by doing so and thank him/her for this. We also made an effort the remove some sections that are not relevant for the presentation.

**Replies to the General comments**

*1/ The text on pages 20-21 (especially lines 544-574) is quite cryptic and inconsistent with the rest of the paper. Here rather than documenting the particular physics package it looks like some ideas or guidelines are presented. They are however hardly justified by some published evidence or experience from other NWP centers. The ACRANEB2 radiation is used without a word of introduction there. The explanation is jumping from microphysics to convection. The presented text is not self-explanatory: What is meant for example "modularity at the level of processes"? I suppose every microphysics scheme is in a way modular by describing separately process by process. Some extensive revision and shortening of this text resulting ideally in description on what has been actually implemented would be highly desirable. (Without too much arguing WHY: This should have been published already elsewhere.)*

Indeed, this part is now rewritten, (see lines 575-629).

*2/ The text has the ambition to serve as an reference (for the ALADIN and also external community). To help this purpose it would greatly improve the navigation when the schemes described in section 2.2 are highlighted (in bold or italic) near the area they are described. When they are mentioned later on in the text a general reader would easily search for them even working with printed text. (For example by highlighting the SSDFI at L276 you will improve the readability of the text at L777.) This could be seen as a poor man's glossary.*

Good suggestion, in the revised manuscript we introduce subsubsections with titles that include the name of the schemes.

We have moved the paragraph that identifies the differentiation of the LAM specific features w.r.t. to the global model features, to improve the structure of the section (now to be found in lines 221-226).

*3/ Some part of the text looks like fillers. A reader may not see a clear reason for incorporating those into the paper. For example Figure 7 brings no specific extra information. It would make perhaps more sense if there is a comparison of the new and old Polish domains. But the same could be summarized easily without a figure. Especially when there is already Figure 6 illustrating roughly the same. Instead of complicated and sometimes very case specific results from newer model versions one would ex pect to find some highlight parameters of scores from global models (those used for boundary conditions), reference LAM version and new LAM version. This is clearly missing.*

Indeed, and section 4 is mostly problematic for this.

We have restructured the text of this section in two clearly distinct subsections: **4.1 Current status of the implementations** and **4.2 Added value.** By doing so we removed the repetitions in the text. Added value is addressed in 4.2 in terms increased realism and by few scores targeting extreme precipitation (comparing the LAM to the global models ARPEGE and ECMWF). It is impossible to give an overview of all of the verifications of the applications in all of the 16 ALADIN countries. We select here a only few cases: Météo-France, ZAMG, RMI and Croatia. We also removed the example of Poland, it is indeed a filler. The other examples are now functional in our opinion.

*There are also plenty of references having no relevance to the paper. As the data assimilation is out of the scope of this paper, a reader may ask why there are so many papers referred to this subject? Some other references are definitely not the most appropriate to the presented subject. Please be honest and provide only relevant references to the presented text.*

The ALADIN System is the result of code developments since 1990. So it is rather normal that there is a lot of literature. Also, it describes three physics packages of ALADIN, AROME and ALARO, which increases the cited literature. Nevertheless, we have taken the opportunity to go through the citations and we removed the ones that we think are redundant, specifically the ones you question in your specific comments below.

*4/ On page 5-6 you describe a procedure of a new model version assembling. It is not clear however how the evolution of the global model code is interfering with this. How the decision about what is implemented at the level of the LAM code is taken? Or does it mean everything developed for global model automatically propagates to the LAM? What about some specific global model issues which are not relevant for the LAM community (like specific treatment of poles)? Do you have some general guidelines or those are solved on case to case basis? Is there some experience with the opposite direction code propagation, like a code developed initially for the LAM model has been made available for the global model too? This perhaps rather particular question tries to reveal a bit more about this rather unique duality that the same code is used for LAM and global model communities.*

First of all, this is not a new method, but this is the first time we describe the existing one in a publication. We have added a paragraph describing the general guidelines (called fundamental rules in the text), in lines 130-157.

**Replies to the Specific comments & Technical corrections**

*1/ Purpose of the paper is given twice: p1/10 and once again p4/90-102. Could it be perhaps unified and reduced to just one list?*

Indeed it is given once in the abstract and once in the introduction. We have reworded the scope in the abstract (lines 11-14) and have removed the scope description from the introduction. It is a filler as you mention above.

*2/ Duplicated text describing the paper limitation: See p1/L15-6 and once again p4/L103-4. In this latter case the repetitive text brings just references to additional papers having no relevance to the described CMCs.*

Here also, the introduction elaborates what is announced in the abstract. We have moved the climate application of the CMC to section 4.3 (lines 848 - 850). Data assimilation is an important part of

NWP so it should be mentioned at least. It comes now after the "This paper is organized as follows ..." part (lines 105-107). The references are removed removed from the introduction.

*3/ p5/L132 text mentions a five-step process defining the scientific developments of the ALADIN System. Apparently there is no such description given in the paper (or it is well hidden). What this "five-step process" should be representing then? When it is given somewhere in the text please make it more explicit to be obvious without an extensive search.*

Indeed. In fact this paragraph is very general and we moved it to the consortium description in the introduction (lines 27 - 37). If you read it carefully it contains 5 steps. But this is irrelevant here, so we do not mentioned the five steps anymore.

*4/ p7/L182-5 Could you explain what is the driving force for you to insist on long time steps? Is it the computational efficiency? Or do you have some specific scientific reason for it? (The computational efficiency doesn't need to be necessarily always justified by long time steps.) This claim feels bit like a dogma. But it is not clear for a reader why this is so important here.*

The model has to be run on a large variety of computing platforms. This is now added in a footnote to the text on p. 7.

*5/ p7/L195 Is the best reference for the SI scheme really the given papers from Caluwaerts et al?*

The first one is not relevant. The second was meant as a review. We removed both of them.

*6/ p8/L215 Could you bit develop on this claim relating the 3km threshold and "important" role for the non-hydrostatic dynamics? First, it is not clear how this threshold is defined in terms of model: Are you referring here the grid point distance of the computational mesh, the shortest wave represented by the model or even a size of the smallest fully resolved feature of the model? Second, please specify the "important role". Could you perhaps give some reference to clarify this claim? To the reviewer's knowledge there is no clear agreement on it. One can perhaps find some effects not simulated with hydrostatic dynamics at those scales. But this still doesn't justify the necessity to use non-hydrostatic dynamics there. One can argue that the non-hydrostatic schemes are only essential when it comes to the simulation of the convective effects. Here we are however referring scales bellow ~100m of grid resolution. Finally, the role of "details of the used numerical scheme" is also not very clear here. Do you mean the true resolution given by the particular numerical scheme? Or something else is meant? To conclude: this sentence sounds like referring some common truth. If there is such an evidence, please provide some reference. Alternatively please make this statement less controversial.*

This is a fundamental discussion. It is not the goal of the paper to have that discussion here, but we describe the current practice with the consortium. We have reworded this, see lines 229 – 233.

We have also removed the reference to (1-5 km) in the introduction, it is not mentioned in line 90 of the new manuscript.

*7/ p8/L220 For VFE there is more fundamental reference of Untch and Hortal to be used rather than the one given in the text.*

Untch and Hortal 2004 are the implementations for the hydrostatic dynamical core. Vivoda and Smolikova developed a new VFE scheme for the NH dynamics. This is now reworded in lines 236 – 239.

*8/ p9/L256 When you give the diffusion order, you should also specify the resolution (and/or) truncation. Does it mean all presented configurations from Table 4 are running with this 4th order horizontal diffusion (including 18 km Aladin-NORAF and 1.3 km Arome France)?*

We actually have written that it is "usually" 4-th order.

This is indeed resolution dependent and may vary quite a lot among the various applications in table 5. Also some applications rely more on SLHD than on the Laplacian operator. A detailed description would lead us too far, so we remove this sentence about the 4$^{th}$ order from the text.

*9/ p11/L291-3002 It is nowhere specified how wide the relaxation area is. From the text at p27 it is apparent the number of coupling zone points is varying. How the given values of parameters p (L301) are modified with respect to the changing z? It is quite evident the optimal value of p must be related to the number of points in the coupling zone and model resolution. Can it be precise here?*

This is now explained (319 - 321). For the power p, ALADIN partners use the values from the standard namelists, i.e. the values mentioned in the paper.

*10/ Several places like p12/L354 and p13/L363,364 are using term "dual parallelization". This is not at all very common term. Presumably it is meant mixed or hybrid MPI/OpenMPI parallelization? If so please change it to hybrid parallelization which is more commonly used name.*

Yes. We now use the wording "hybrid (MPI/OpenMPI) parallelization" to clarify that, see line 194 and line 379.

*11/ p16/L448 The sensitivity of the scheme to the time step length has been... changes. This sentence brings no information to a general reader. Please either provide some details or drop it.*

We drop the line.

*12/ p17/L487 This 15 minutes intermittency is used at every Arome configuration? Your example is given with Arome 1.3 km and 50s time-step. But there are some 2.5 km and 90s Arome configurations in the Table 4. Does it mean the 15 minutes remains fixed regardless the actual time-step length?*

Yes, indeed, this 15 minutes choice does not depend of the horizontal resolution nor time step of the model. In Table 4, 90 should refer to vertical levels, not time step. We change "full radiation computations are performed once every 15 min" by "in all AROME configurations (2.5 or 1.3 horizontal resolution) full radiation computations are performed once every 15 min. For intermediate time steps, only solar azimuth angle varies." (lines 515 -518).

*13/ p14-p19 (Arome CMC) At scales of 1.3km is certainly not negligible a contribution of horizontal mixing/turbulence. Please give some details about your representation of those highly non-liner horizontal effects.*

Yes, indeed, at 1.3km scales, there is probably a not negligible contribution of horizontal mixing/turbulence, but due to diffusive processes (Semi-Lagrangian advection for instance), the 'effective' resolution of the model is far from being 1.3km (Ricard et al.,2013 ) and we can still use a 1-D turbulence scheme (Honnert et al., 2016).

Honnert, R. : Representation of the grey zone of turbulence in the atmospheric boundary layer, Adv. Sci. Res., 13, 63-67, https://doi.org/10.5194/asr-13-63-2016, 2016.

Ricard, D., Lac, C., Riette, S., Legrand, R. and Mary, A. (2013), Kinetic energy spectra characteristics of two convection-permitting limited-area models AROME and Meso-NH. Q.J.R. Meteorol. Soc., 139: 1327–1341. doi:10.1002/qj.2025

This discussion would lead us too far so we prefer to not change the text.

*14/ p20/L546-8 Separation of scales is not unnatural. I believe it is meant rather arbitrary. The separation of processes to dry and moist is equally unnatural/arbitrary, by the way. The text is not correct. There can't be such clear separation. This just says the microphysics is called twice in this case.*

We agree. The separation issue has not been very well explained, thank you for pointing it out. We redrafted the relevant part of the text to make it clear we speak about diverging parameterization concepts. We also clarify there is a single (and not double) call to the microphysics, see lines 586 - 594.

*15/ p22/23 Could you specify the closure used for the turbulence scheme? Is it closed by a mixing length? And if so, which one?*

We now provide an explanation: lines 669 – 681.

*16/ p23/L622-626 Rather strange text with a link to turbulence but then mentioning microphysics. What is the relevance of it? Does the microphysics influences the turbulence?*

The paragraph is now rewritten: 690 – 694.

*17/ p24/L654-5 This is rather strong claim. Could you perhaps give some reference or bring some more evidence supporting it?*

This sentence is now rewritten in lines 723 – 726.

*18/ p22/24 Can you give some description for the microphysics and gravity wave drag parameterization? A reader may wonder what makes those two schemes so unattractive that the only information about them can be found in the table 3.*

It is the Catry et al. 2008 GWD. This is now introduced in a bit more detail in the new version of the text (it is the same as the one for the ALADIN CMC), see lines 727 – 732.

*19/ The paper of Lopez(2002) being referred as the microphysics description is introducing only three prognostic variables: water vapour, cloud condensates and falling precipitation. Is this really the case for the presently used microphysics? If not could you explain the choice of prognostic variables related to the microphysics in ALARO CMC?*

This has been addressed in the new text that was provided to respond to your general comment 1, see line 626.

*20/ p25/L671: Missing "with" or "to"?*

Indeed. This is now corrected, line 755.

*21/ p26/Table 4. Please specify the date of validity. The actual state could be evolving.*

Good point. A priori it is the date of submission, but it helps if it is added to the caption of the table. We also added it to the figure of the domains.

*22/ p28/Fig 8 Are the curves based on annual verification of the two models? If so it is truly impressive, but better to say it more explicitly. In the other case please specify the verification period. It would be also useful to add the zero horizontal line (especially to the upper panel) in order to help the results interpretation.*

This part of the text was removed in reply to your above comment on fillers.

*23/ p32/Fig 11 The red dot is nearly invisible (especially when printed). Please use some better way to highlight it. This figure demonstrates the superiority of the newer version of ALARO over the operational one. Could you then add the operational results to illustrate it graphically?*

The red dots are now replaced by black dots.

The difference in scores between the 1.3 km and our current 4 km resolution operational version is minor. This figure illustrates the increase in realism when increasing the resolution. It would lead too far to provide a case study.

*24/ p33/Fig 12 Could you please zoom the figure to its lower third? It is really difficult to follow the presented the multiple lines of HR-alaro-88 and HR-alaro-HRDA.*

Indeed. This is now done.

**Reply to the comments of reviewer 2**

We thank Per Unden for his comments and suggestions. We have implemented his suggestions in a revised manuscript.

**Replies to his detailed comments**

1. *typo: line 273 p 10: In operation . . .. should be operational*

This is corrected in the new version of the manuscript, see line 295.

2. *line 240 p 9 : . . .. more conservative semi-Lagrangian . . ... : please make the link to the same but a bit longer explanation of this scheme around line 413. Perhaps also here refer to it as COMAD to make it consistent.*

The description more extensive description of COMA mentioned in the AROME CMC part is moved to section 2.2, and lines 439-440 then refer to section 2.2.1. This should improve readability.

3. *line 420 p 11 : Please make the comparison with the TKE scheme in ALADIN/ARPEGE on line 391. From the text it appears to be the same scheme albeit with some different variables but it is relevant here to state what is shared and what the differences are between the TKE schemes, or indeed if they are or could be the same or share the same code.*

The turbulence scheme used in Arome differs from the one used in Arpege/Aladin mainly on the vertical discretization of TKE defined on full levels versus half levels respectively. Both schemes have been compared in several 1D cases and the results are very similar. There is an ongoing work to share exactly the same code.

This is now explained in the text in lines 447 – 450.

4. *line 439, p 16: In this way . . .. of a RH-scheme . . .. : I don't understand this at all. The earlier sentences all give the message that the scheme is everything but a RH scheme! Which of the "ways" just mentioned makes it a RH scheme? Please qualify and explain or change if it is an error.*

We would say that in such particular conditions (no turbulence), with this extra term, the cloud schemes acts as a RH-Scheme. We explain this now in lines 465 – 469: "In order to represent ..."

5. *line 450 p 16 2-moment scheme . . .. implemented . . .. : please add something like not activated since on 441 you describe the current one moment scheme, confusing for the non-initiated.*

We added in the text "(used in research mode, not yet activated in operational)" in line 480-481.

6. *Line 473 p 16: Again, please compare with ALADIN radiation on line 388. There are many common components in the basic scheme it seems.*

ALADIN and AROME used radiation schemes are the same (RRTM for LW and Fouquar Morcrette for SW). There are only small differences in terms of cloud overlap assumptions and calling frequency (1h in ALADIN versus 15' in AROME).

The text has been modified to state this in line 501.

*7. typo line 499 : Météo . . . - missing*

This is now corrected.

*8. Before Table 2. There should be a Table for the ALADIN baseline CMC as well – to be able to compare AROME and ALARO!*

Indeed. The table is now added.

*9. Figure 8. Please state if it is for the whole year of 2013 or which period.*

This figure has been removed in reply to the general comment 3 of reviewer 1.

[revised manuscript text omitted]

---

## Author Response (AR2)

Dear Editor,

We have addressed your comments and requests, see below.

**Detailed replies**

*- The font size for figure 3 should be increased to be more consistent with the manuscript text. At present the axis and tickmark labels are difficult to read.*

This is done.

*- Contour labels should be added to figure 5 directly, rather than relying on caption text to identify these labels.*

This is done.

*- I have great difficulty reading Figure 6, largely because the realistic land cover coloring does not provide sufficiently high contrast with the domains. I would like to ask that the authors consider how this figure can be adjusted to improve readability.*

We have:
- made the background of the globe less prominent,
- increased the contrast of the lines of the different limited areas,
- removed the background color of the legend to the left of the figure.

We believe the readability improved by this. We use this figure frequently for our reporting within the consortium and will use this one in the future.

*- Regarding Figure 10, the color scheme for the axis box, labels, and text should be changed to black to improve the contrast of the figure against the background.*

This is done.

Thanks for the suggestions.

Best regards,

Piet Termonia